# Op-CAD: Benchmarking and Investigating Operation-oriented CAD Generation

**Yixue Bai** [1]   **Yufei Gu** [1]   **Zeke Xie** [1]

## Abstract

Recent research has made growing efforts to leverage large language models (LLMs) for computer-aided design (CAD), a domain that demands advanced geometric and spatial reasoning across long operation sequence. However, existing studies remain limited in addressing complex modeling tasks that necessitate step-by-step reasoning, primarily due to the scarcity of high-quality CAD datasets and the absence of fine-grained evaluation frameworks. In response to these challenges, we introduce Op-CAD, the first large-scale, multimodal dataset for operation-oriented CAD generation, encompassing four operation types and five modalities. Furthermore, we introduce a novel CAD parsing module together with a geometry-guided hierarchical annotation pipeline, which decomposes modeling sequences into discrete operations and substantially improves the annotation accuracy of Vision-Language Models (VLMs). Based on our dataset, we redefine the CAD modeling task by decoupling geometric and spatial perspectives and introduce a novel metric, Chamfer/Fillet Intersection over Union (CF-IoU), to fill the void in assessing chamfer and fillet operations. By comprehensively evaluating eight LLMs on Op-CAD, we establish a benchmark for current models on operation-oriented tasks. Finally, we investigate performance enhancement strategies through fine-tuning on Op-CAD and propose Chain-of-Operation (COOP), a novel prompting strategy that emulates human-engineer reasoning. Our project page is available at https://baiyixue01.github.io/op-cad/.

## 1. Introduction

Computer-Aided Design (CAD) has long been an indispensable tool for engineers, enabling the creation of precise, para-

[1]The Hong Kong University of Science and Technology (Guangzhou), Guangzhou, China. Correspondence to: Zeke Xie <zekexie@hkust-gz.edu.cn>.

*Proceedings of the 43rd International Conference on Machine Learning*, Seoul, South Korea. PMLR 306, 2026. Copyright 2026 by the author(s).

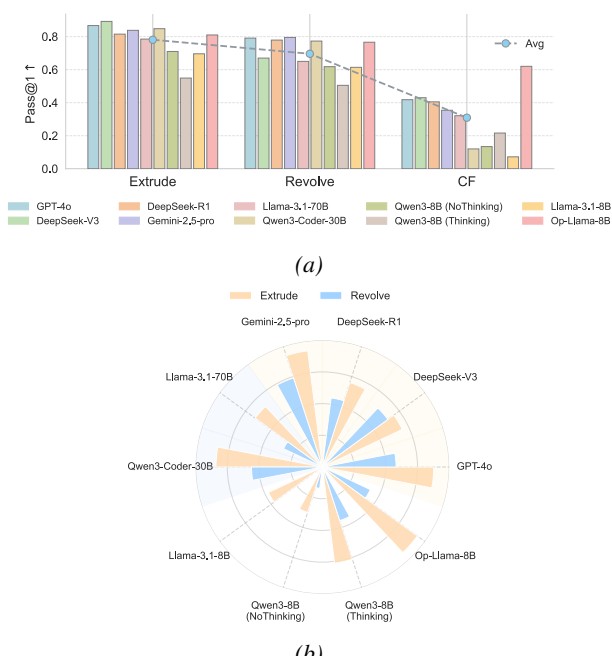

*(a)*

*(b)*

*Figure 1.* Comparison of model performance across different operations. Op-Llama-8B, the Llama-3.1-8B fine-tuned on Op-CAD, is largely enhanced by our dataset. (a) Model success rates (Pass@1), where *Avg* denotes the mean Pass@1 across all models within each operation group. **CF** refers to two advanced operations, namely Chamfer and Fillet. (b) 3D geometric metric (Mean CD), where Mean CD is normalized and reversed so that higher is better.

metric models that underpin modern manufacturing (Sarcar et al., 2008). As Large Language Models (LLMs) advance, using LLMs for CAD generation has attracted increasing attention and achieved substantial progress. CAD modeling is a sequence-based task that demands intensive spatial and geometric reasoning. Each modeling step must be properly composed into a long sequence by completing two key sub-tasks: (1) inferring the geometric shape and (2) determining its spatial location(Yuan et al., 2025), which naturally suggests that a step-by-step formulation is a promising way to tackle complex CAD modeling. However, although LLMs have demonstrated strong problem-solving ability by decomposing tasks and performing step-by-step reasoning (e.g., in mathematics)(Wei et al., 2022), a similar insight—constructing complex CAD models in an operation-level, step-by-step manner—has not yet clearly emerged in

LLM-based CAD generation. This gap is primarily driven by two factors.

First, the scarcity of suitable datasets. Existing datasets show limitation in three aspects. **(1)** Lack of High-Complexity Models. The evolution of any field relies on increasingly challenging tasks to push its functional boundaries. However, models in existing CAD datasets lack sufficient complexity in both sequence depth and operational diversity. As the most widely used benchmark, DeepCAD (Wu et al., 2021) exhibits an average sequence length of only 2 and is largely restricted to basic extrusion operations. Consequently, competitive performance can be easily attained using simplistic frameworks, which inadvertently diminishes the incentive to explore more sophisticated, fine-grained decomposition methods. **(2)** Insufficient Parsing Granularity. The granularity of data parsing is fundamental to enabling operation-level modeling exploration. Current datasets primarily offer high-level modeling histories but lack explicit step-by-step decomposition and detailed parsing of cross-operation relationships. Coupled with limited data modalities, these deficiencies further constrain the potential for operation-oriented CAD generation. **(3)** Inaccurate Fine-Grained Annotation. Fine-grained operations require precise labeling to avoid error accumulation. While VLMs are widely used for image-based annotation, they struggle to infer the underlying geometry and topology of CAD models from raw pixels. This leads to low-precision outputs that cannot support the rigorous demands of operation-oriented CAD generation.

Second, the lack of a comprehensive evaluation system capable of providing multi-faceted and in-depth analysis of model performance. Specifically, this deficiency is manifested in three critical dimensions: **(1)** Coarse Task Definitions. As the foundation for systematic study, the problem formulation in most current LLM-based CAD generation research focuses on generating an entire model in a single shot, lacking multi-level task definitions that decompose the problem into well-specified sub-tasks. **(2)** Lack of Decoupled Geometric and Spatial Evaluation. Current CAD evaluation methods focus primarily on the final output, thereby masking the intermediate modeling errors and hindering a deeper analysis of the underlying construction mechanisms. **(3)** Missing Metrics for Advanced Operations. A significant gap in evaluation metrics remains for edge-based operations, such as fillet and chamfer. Consequently, despite being ubiquitous in industrial design, these operations are frequently overlooked in current CAD generation research.

The limitations summarized above make it difficult to achieve the complexity and fidelity required for real-world CAD applications. To address these critical gaps and facilitate a high-level architectural advancement in LLM-based CAD, we present Op-CAD, which overcomes these bottle-necks through the following four key contributions:

- **The First Operation-Oriented Multi-modal CAD Dataset:** We construct Op-CAD dataset, comprising 128k high-quality operation instances across four operation types and five data modalities. To the best of our knowledge, Op-CAD is the most modality-rich and operationally diverse large-scale CAD dataset currently available. Notably, the average sequence length in Op-CAD is eight times greater than that of the widely used DeepCAD benchmark, providing significantly higher complexity for modeling tasks.

- **Novel CAD Model Parsing and Annotation Pipeline:** We design a CAD parsing module and a geometric-guided hierarchical data annotation pipeline. We are the first to explicitly decompose CAD modeling sequences into fine-grained operations and topological relationships. By innovatively leveraging precise geometric data to guide VLMs, our pipeline significantly enhances CAD semantic annotation.

- **Comprehensive and Hierarchical Evaluation:** We are the first to define the CAD modeling task from the perspective of geometric and spatial reasoning. Specifically, we propose a novel metric, Chamfer/Fillet Intersection over Union (CF-IoU), to address the void in evaluating these advanced operations. By benchmarking eight LLMs on the Op-CAD task, we reveal significant spatial reasoning limitations in current models and provide an in-depth analysis for future research.

- **Investigating Strategies for Enhancing CAD Modeling:** We design two naturally resulted strategies, including a novel Chain-of-Operation (COOP) prompting strategy and Op-CAD-based tuning/training. COOP facilitates human-like spatial reasoning without additional training. The resulted Op-Llama-8B, a Llama-3.1-8B model fine-tuned on Op-CAD, can even outperform 8 times larger Llama-3.1-70B.

**Conflict of Interest Disclosure.** The authors declare no financial conflicts of interest.

## 2. Related Work

**CAD Dataset.** Early efforts contributed large-scale B-Rep repositories such as ABC (Koch et al., 2019) and CC3D-Ops (Dupont et al., 2022). However, B-Rep datasets typically lack explicit modeling histories, making it hard to recover the underlying construction process (Heidari & Iosifidis, 2025). Subsequent works therefore parse CAD models into procedural command sequences (Zhou et al., 2023; Willis et al., 2020), with DeepCAD (Wu et al., 2021) becoming a widely used benchmark (179k models). More recently, cross-modal CAD generation has augmented datasets with images (You et al., 2024; Chen et al., 2025), text (Khan et al.,

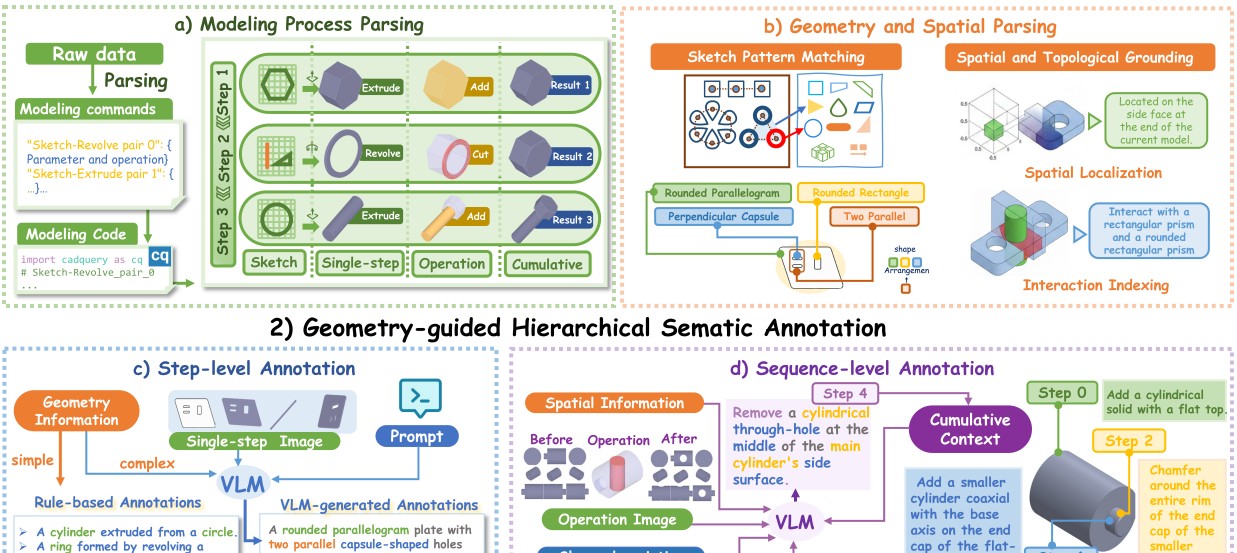

*Figure 2.* Overview of our three-stage CAD data annotation pipeline. Stage I parses and renders raw CAD data into valid operations and multi-level visual representations. Stages II and III leverage Vision-Language Models (VLMs) to annotate each operation instance with natural language descriptions of geometric shape and positional context. By integrating a CAD parsing module with generative models, the pipeline minimizes human annotation effort and produces a high-quality, multi-modal CAD dataset.

2024; Xu et al., 2024), and code (He et al., 2025; Niu et al., 2025; Badagabettu et al., 2024). Despite this progress, textual annotations remain limited by general-purpose VLMs' weak spatial reasoning and CAD-specific knowledge (Heidari & Iosifidis, 2025). Additionally, existing open-source datasets for LLM-based CAD modeling are largely limited to simple geometries and a narrow operation set (often extrusion), and they pay limited attention to inter-step topological dependencies across the modeling chain, which constrains both operation coverage and supervision for CAD-specific reasoning (Xie & Ju, 2025; Grattafiori et al., 2024; Guo et al., 2022; Ma et al., 2025).

**LLM-based CAD Modeling.** Recently, LLMs have shown strong engineering understanding capabilities(Doris et al., 2025; Jadhav & Farimani, 2024) and achieved promising results on text-driven CAD modeling. LLM4CAD(Li et al., 2024) and Query2CAD(Badagabettu et al., 2024) generate programmatic CAD scripts using LLMs, while CAD-MLLM(Xu et al., 2024) produces CAD command sequences conditioned on text. CAD-Llama(Li et al., 2025b) trains an LLM to generate structured CAD code, and ReCAD(Li et al., 2025a) trains a vision–language model for text-to-CAD via reinforcement learning. Beyond CAD generation, several efforts have also investigated CAD editing. Flexcad(Zhang et al., 2024) enables region-specified CAD edits with LLMs, and CAD-Editor(Yuan et al., 2025) decomposes CAD editing into locating and modifying according to textual instructions. However, most existing LLM-based

text-to-CAD studies focus on high-level, one-shot generation of a single CAD model (Mustapha, 2025), leaving operation-oriented, step-by-step tasks underexplored. This gap substantially limits their ability to handle more complex CAD modeling problems with longer operation sequences.

## 3. Op-CAD Dataset

Op-CAD dataset contains 128k high-quality CAD operation pairs spanning four types of CAD operations. Each pair is provided in two complementary views: a step-level view focusing on the current operation and its immediate geometry, and a sequence-level view that situates the operation within the full procedural history. For each view, we provide the corresponding B-Rep geometry, textual description, modeling command, executable CadQuery code, and multi-view renderings. Furthermore, each instance is annotated with fine-grained inter-step metadata parsed using a CAD parsing module. In this section, we first introduce our novel CAD parsing module (Section 3.1). Then, we present the geometry-guided, hierarchical data annotation pipeline (Section 3.2). Finally, we showcase the statistics of Op-CAD Dataset (Section 3.3).

### 3.1. CAD Parsing Module

To bridge the gap between low-level CAD operations and high-level linguistic descriptions, we design a Geometric Parsing Model, which consists of two interconnected compo-

*Table 1.* Comparison of CAD datasets across modalities, operations, and parser levels. We categorize parsing granularity into three levels: **Seq.** (full operation sequence), **Step** (individual modeling features and parameters), and **Relation** (inter-operation dependencies and topological constraints). Compared to existing Dataset, Op-CAD supports the most diverse modalities and operation types, providing comprehensive multi-level parsing data.

| Dataset | Size | Modalities | | | | | 3D Operations | | | | Model Parsing Level |
|---|---|---|---|---|---|---|---|---|---|---|---|
| | | Geometry | Code | Cmd Seq. | Image | Text | Extrude | Revolve | Chamfer | Fillet | |
| ABC (Koch et al., 2019) | 1M+ | ✓ | ✗ | ✗ | ✗ | ✗ | – | – | – | – | Seq. |
| CADParser (Zhou et al., 2023) | 40K+ | ✗ | ✗ | ✓ | ✗ | ✗ | ✓ | ✓ | ✓ | ✓ | Seq. |
| DeepCAD (Wu et al., 2021) | 179K | ✗ | ✗ | ✓ | ✗ | ✗ | ✓ | ✗ | ✗ | ✗ | Seq. |
| Fusion360 Rec. (Willis et al., 2020) | 8.6K | ✓ | ✗ | ✓ | ✗ | ✗ | ✓ | ✗ | ✗ | ✗ | Seq. |
| Img2CAD (You et al., 2024) | 4.6K | ✗ | ✗ | ✓ | ✓ | ✗ | ✓ | ✗ | ✗ | ✗ | Seq. & Step |
| Text2CAD (Khan et al., 2024) | 158K+ | ✗ | ✗ | ✓ | ✗ | ✓ | ✓ | ✗ | ✗ | ✗ | Seq. |
| Omni-CAD (Xu et al., 2024) | 453K | ✓ | ✗ | ✓ | ✓ | ✓ | ✓ | ✗ | ✗ | ✗ | Seq. |
| CAD-Editor (Yuan et al., 2025) | 122K+ | ✗ | ✗ | ✓ | ✗ | ✓ | ✓ | ✗ | ✗ | ✗ | Step |
| GenCAD-Code (He et al., 2025) | 163K+ | ✗ | ✓ | ✗ | ✓ | ✗ | ✓ | ✗ | ✗ | ✗ | Seq. |
| **Op-CAD (Ours)** | 128K+ | ✓ | ✓ | ✓ | ✓ | ✓ | ✓ | ✓ | ✓ | ✓ | Seq. & Step & Relation |

*Table 2.* Formal definition of geometric and spatial priors $\mathcal{P}_j$.

| Component | Symbol | Definition and Semantic Meaning |
|---|---|---|
| Geometric $G_j$ | $\phi$ | Profile shape category, $\phi \in \Phi$ (*canonical primitives* or *unrecognized*). |
| | $\alpha$ | Arrangement pattern, $\alpha \in \mathcal{A}$ (*regular patterns* or *unrecognized*). |
| | $\kappa$ | Primitive count, $\kappa \in \mathbb{Z}^+$. |
| Spatial $\mathcal{S}_j$ | $\mathbf{p}$ | Semantic position, $\mathbf{p} \in \{\text{mid, end, near\_end}\}^3$ relative to global BBox. |
| | $\sigma$ | Target surface type, $\sigma \in \Sigma$ (*end cap* or *side surface*). |
| Topological $\mathcal{T}_j$ | $\mathcal{I}$ | Interaction set, $\mathcal{I} = \{i \mid i < j, \text{Intersects}(t_i, t_j) \neq \emptyset\}$. |

nents: a Modeling Process Parser, followed by a Geometry and Spatial Parser, which are showed in Figure 2.1).

**Modeling Process Parser.** A raw CAD model can be represented as a sequence of historical operation $\{t_1, t_2, \ldots, t_n\}$. In their raw command sequence form, various type of parameters are often entangled, making it difficult to extract clear semantic intent. To resolve this, we decompose each raw operation $t_j$ into a structured triplet $O_j = \{g_j, \omega_j, \mathcal{C}_j\}$:

- Geometric shape of singe step $g_j$: The isolated 3D shape generated by the current sketch-operation pair.
- Topological Interaction $\omega_j$: Includes (i) *Boolean relationships* (e.g., Join, Cut) for constructive geometries, and (ii) *Topological refinements* (e.g., Fillet, Chamfer) that modify existing boundary edges.
- Cumulative Prior Operations $\mathcal{C}_j$: The cumulative modeling state up to step $j$, corresponding to the sequence of prior operations $\{t_1, t_2, \ldots, t_{j-1}\}$.

As illustrated in Figure 2 a), these structured triplets $O_j$ serve as the foundational representation to derive multiple data modalities, including step-level and sequence-level command sequences, executable code, B-Rep geometries, and multi-view renderings.

**Geometry and Spatial Parser.** Building upon the parsed modeling information $O_j$, we extract a set of geometric and spatial priors $\mathcal{P}_j = \{G_j, \mathcal{S}_j, \mathcal{T}_j\}$. As formally defined in Table 2, $G_j$ represents the sketch-derived geometric shapes, $\mathcal{S}_j$ denotes the spatial localization within the global model, and $\mathcal{T}_j$ indicates the indices of previously interacted operations. Collectively, these priors provide inter-step metadata for each modeling operation, deconstructing the deep geometric and spatial information hidden within the CAD modeling sequence that is difficult to infer directly.

### 3.2. Geometry-guided Hierarchical Annotation

We adopt a hierarchical annotation strategy inspired by CAD-Llama (Li et al., 2025b) to obtain detailed and structured textual descriptions of CAD operations using Gemini-2.5-Pro. To reduce VLM hallucinations and improve annotation precision, we leverage the geometric cues $O_j$ and the inter-step metadata $\mathcal{P}_j = \{G_j, \mathcal{S}_j, \mathcal{T}_j\}$ parsed by our CAD parsing module as explicit guidance. As shown in Figure. 2 2), the annotation pipeline is divided into step-level and sequence-level stages.

**Step-level Annotation.** In this stage, a description of the geometry created in the current step is generated. To maximize precision, we adopt a hybrid strategy conditioned on the profile shape category $\phi \in G_j$. Let $\Phi_c \subseteq \Phi$ denote the set of canonical primitives recognizable by the CAD parsing module. The final step-level description $A_{step,j}$ is synthesized as:

$$A_{step,j} = \begin{cases} \mathcal{R}(G_j) & \text{if } \phi \in \Phi_c \\ \text{VLM}(\mathcal{I}_{step,j}, g_j, G_j, \text{prompt}_1) & \text{otherwise} \end{cases}$$
(1)

where $\mathcal{R}(\cdot)$ is a rule-based function that maps recognized geometric shapes into structured textual descriptions using a predefined template. For unrecognized or complex composite shapes, we prompt the VLM to generate shape

**Algorithm 1** Salient Context Window Construction

---

**Require:** step $j$, capacity $K$, interactions $\mathcal{I}_j$, op types $o_j$, history
$\quad\quad \mathcal{H}_{\text{full}} = \{A_{seq,1}, \dots, A_{seq,j-1}\}$
**Ensure:** $\mathcal{H}_j$
1: **if** $j - 1 \leq K$ **then**
2: $\quad$ $\mathcal{H}_j \leftarrow \mathcal{H}_{\text{full}}$ $\quad\quad\quad\quad\quad\quad$ ▷ use all history
3: **else**
4: $\quad$ **if** $|\mathcal{T}_j| \geq K$ **then**
5: $\quad\quad$ $i \leftarrow \min(\mathcal{T}_j)$
6: $\quad\quad$ $\mathcal{H}_j \leftarrow \{A_{seq,i}, \dots, A_{seq,i+K-1}\}$
7: $\quad$ **else**
8: $\quad\quad$ $\mathcal{H}_j \leftarrow \{A_{seq,t} \mid t \in \mathcal{T}_j\}$
9: $\quad\quad$ **for** $p \leftarrow j-1$ **down to** 1 **do**
10: $\quad\quad\quad$ **if** $|\mathcal{H}_j| = K$ **then**
11: $\quad\quad\quad\quad$ **break**
12: $\quad\quad\quad$ **end if**
13: $\quad\quad\quad$ **if** $p \notin \mathcal{T}_j$ and $\omega_p \notin \{\text{Fillet}, \text{Chamfer}\}$ **then**
14: $\quad\quad\quad\quad$ $\mathcal{H}_j \leftarrow \mathcal{H}_j \cup \{A_{seq,p}\}$
15: $\quad\quad\quad$ **end if**
16: $\quad\quad$ **end for**
17: $\quad\quad$ $\mathcal{H}_j \leftarrow \text{SortByIndex}(\mathcal{H}_j)$
18: $\quad$ **end if**
19: **end if**

---

descriptions conditioned on the multi-view shape image set $\mathcal{I}step, j$, the operation type inferred from the single-step geometry $g_j$, and the recognized sub-profile shape types and counts encoded in $G_j$, prompt1 denotes a step-level structured prompt.

**Sequence-level Annotation.** This stage generates sequence-level annotations $A_{seq,j}$ that connect individual operations into a coherent modeling chain. Since certain spatial placement semantics (e.g., centered vs. edge placement or through-holes) are difficult to infer from images alone, in addition to the sequence-level image set $\mathcal{I}_{seq,j}$ and sequence-level structured prompt$_2$, the VLM is prompted with the step-level shape description $A_{step,j}$, spatial cues $\mathcal{S}_j$, and topological interactions $\mathcal{T}_j$.

Notably, we further provide a cumulative context $\mathcal{H}_j$ to supply modeling history. To mitigate attention decay in long sequences and emphasize the most relevant prior operations, we construct $\mathcal{H}_j$ as a Salient Context Window with a fixed capacity $K = 5$, as detailed in Algorithm 1. The final sequence-level description $A_{seq,j}$ is defined as:

$$A_{seq,j} = \text{VLM}(\mathcal{I}_{seq,j}, A_{step,j}, \mathcal{S}_j, \mathcal{T}_j, \mathcal{H}_j, \text{prompt}_2) \quad (2)$$

### 3.3. Op-CAD Statistics

Op-CAD dataset comprises 128,449 modeling operation instances curated from over 27k unique professional mechanical CAD models. It encompasses five data modalities across two distinct dimensions, supplemented by rich metadata that captures the intricate geometric and spatial modeling information for each operation. We summarize detailed Op-CAD statistics and compare it against existing open-source CAD datasets, highlighting advances in modality coverage,

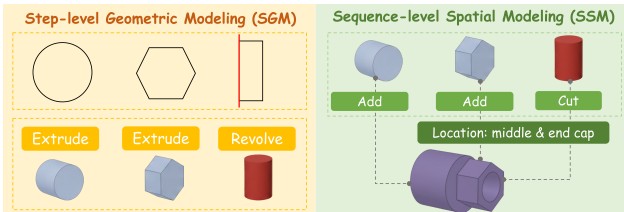

*Figure 3.* Illustration of our task definitions.

operation diversity, and parsing granularity, as detailed in Tables 3 and Tables 1.

Comparing to the widely used DeepCAD dataset, Op-CAD not only introduces a broader range of modeling operations but also features significantly higher complexity and a long-tail distribution of sequence lengths. As demonstrated by the comparative analysis in Table 4 and Table 6.

*Table 3.* Op-CAD statistics.

| Statistics | Counts (Percentage) |
|---|---|
| **Total data** | 128449 (100%) |
| – Sketch-Extrude | 82911 (64.55%) |
| – Sketch-Revolve | 35380 (27.54%) |
| – Chamfer | 4656 (3.63%) |
| – Fillet | 5502 (4.28%) |
| **Data Modalities** | 5 |

*Table 4.* Comparison of operation counts per model among Op-CAD, DeepCAD, and Omni-CAD. Op-CAD contains significantly longer operation sequences, with an average length over 8 times that of DeepCAD and over 6 times that of Omni-CAD, reflecting increased modeling complexity.

| Dataset | Avg | Max | [0–3] | [4–20] | [21–50] | [51–105] | [106–277] |
|---|---|---|---|---|---|---|---|
| Op-CAD | 14.29 | 277 | 32.42% | 50.48% | 9.50% | 5.86% | 1.74% |
| DeepCAD | 1.73 | 102 | 92.23% | 7.71% | 0.05% | 0.0042% | 0.00% |
| Omni-CAD | 2.24 | 102 | 85.67% | 14.06% | 0.21% | 0.06% | 0.00% |

## 4. Op-CAD Evaluation System

Based on the Op-CAD dataset, we design a comprehensive evaluation framework to assess LLMs' CAD modeling capabilities, covering step-level geometric accuracy and sequence-level spatial reasoning. In this section, we first present the geometric and spatial task definitions (Section 4.1). Next, we introduce the evaluation metrics, including a novel metric tailored to chamfer and fillet operations (Section 4.2). Finally, we describe our hierarchical evaluation strategy (Section 4.3).

### 4.1. Task Definitions

Given the exceptional programmatic capabilities of LLMs and the requirements for zero-shot evaluation, we adopt

CadQuery as the core modeling format. Task definition overview is illustrated in Figure 3.

**Step-level Geometric Modeling (SGM).** This task requires the model to map a textual instruction $A_{s,j}$ into an executable modeling script $s_{step,j}$. Formally, this is represented as:

$$s_{step,j} = \text{LLM}(A_{s,j}) \qquad (3)$$

To construct the final geometry, the model must perform a programmatic synthesis that transitions from sketching to 3D operations:

$$s_{step,j} \Rightarrow \{\text{Sketch}, \text{Op}\} \qquad (4)$$

Here, Sketch represents the 2D profile construction involving primitives such as lines, arcs, and circles, while Op denotes the 3D modeling operations—specifically extrusion or revolution—that transform the 2D sketch into the final 3D geometry.

**Sequence-level Spatial Modeling (SSM).** This task evaluates the model's spatial reasoning within a cumulative modeling chain. Given the preceding modeling script sequence $\mathcal{S}_{<j}$ and a sequence-level spatial instruction $A_{seq,j}$, the model is required to generate the subsequent script $s_{seq,j}$. This process is formalized as:

$$s_{seq,j} = \text{LLM}(A_{seq,j}, \mathcal{S}_{<j}) \qquad (5)$$

where $s_{seq,j}$ must resolve:

$$s_{seq,j} \Rightarrow \{\text{Localization}, \text{Interaction}\} \qquad (6)$$

Here, localization involves selecting the appropriate workplane based on prior geometry. Interaction specifies the required topological relationship with existing features according to the instruction, encompassing both boolean operations (cut or join) and local refinements (chamfer or fillet).

### 4.2. Metrics

To assess the accuracy and robustness of LLMs in script-based modeling, we employ the *Pass@k* metric ($k = 1, 2$) to measure the execution success rate (Li et al., 2025c). The geometric alignment between the ground truth and the generated models is measured using *Chamfer Distance (CD)* and *Hausdorff Distance (HD)*.

However, current metrics fail to evaluate the accuracy of chamfer and fillet operations. Existing 3D evaluation methods, such as point-cloud-based CD/HD or volumetric-based Intersection over Ground Truth (IoGT)(Alrashedy et al., 2024), are inherently biased toward global structural fidelity. However, these metrics fail to capture localized refinements such as chamfers or fillets, as the geometric variations induced by these operations are often insignificant relative to

the overall model volume. To overcome this insensitivity, we design a novel metric, *Chamfer/Fillet Intersection over Union (CF-IoU)*.

**CF-IoU** measures the discrepancy between targeted edges of chamfer or fillet operations in the ground-truth model and those in the generated result. Specifically, we extract the exact sets of edges that are chamfered or filleted from both the ground-truth and generated models using the CadQuery library, and then quantify their overlap, defined as:

$$\text{CF-IoU} = \frac{|E_{\text{pred}}^{\text{CF}} \cap E_{\text{gt}}^{\text{CF}}|}{|E_{\text{pred}}^{\text{CF}} \cup E_{\text{gt}}^{\text{CF}}|}, \qquad (7)$$

where $E_{\text{pred}}^{\text{CF}}$ and $E_{\text{gt}}^{\text{CF}}$ denote the sets of edges affected by chamfer or fillet operations in the predicted and ground-truth models, respectively.

### 4.3. Hierarchical Evaluation Strategy

We perform a hierarchical evaluation at both the step and sequence levels to comprehensively analyze the model's geometric and spatial reasoning capabilities in CAD modeling.

Given the generated step-level script $s_{\text{step},j}$ and sequence-level script $s_{\text{seq},j}$, we first execute $s_{\text{step},j}$ to generate the geometric model $M_{\text{step},j}$. Then, we combine the sequence-level script $s_{\text{seq},j}$ with $s_{\text{step},j}$, and execute them together with the history modeling script sequence $\mathcal{S}_{<j}$ to obtain the sequence-level cumulative model $M_{\text{seq},j}$. This process can be represented as:

$$\begin{aligned} M_{step,j} &= \text{exec}_1(s_{step,j}) \\ M_{seq,j} &= \text{exec}_2(\mathcal{S}_{<j}, s_{step,j}, s_{seq,j}) \end{aligned} \qquad (8)$$

To disentangle geometric construction from spatial reasoning capabilities, we evaluate the step-level geometric model $M_{step,j}$ and the sequence-level cumulative model $M_{seq,j}$ separately. For each level, we report the Pass@k execution success rate and quantitative 3D metrics (CD, HD).

## 5. Experiments

In this section, we first introduce the experimental setup (Section 5.1). We then comprehensively compare the capabilities of various models on Op-CAD and summarize key findings (Section 5.2). Finally, we investigate several enhancement strategies, including fine-tuning on Op-CAD, one-shot prompting, and our proposed Chain-of-Operation (COOP) prompting (Section 5.3).

### 5.1. Experimental Setup

**Data Settings.** We collected 27,281 unique professional CAD models from the dataset provided by CADParser

*Table 5.* Evaluation of large models on Op-CAD. SGM denotes the Step-level Geometric Modeling task, and SSM denotes the Sequence-level Spatial Modeling task. We report Pass@1/Pass@2 as success rates, and use Chamfer Distance (CD) and Hausdorff Distance (HD) to measure geometric precision, where values are scaled by $10^3$ for readability. CF-IoU is used to evaluate the accuracy of chamfer and fillet operations. Relative improvements over corresponding base models are shown in parentheses; improvements are highlighted in red, while drops are highlighted in green.

| Model | SGM | | | | | SSM | | | | | |
|---|---|---|---|---|---|---|---|---|---|---|---|
| | Pass@1 ↑ | Pass@2 ↑ | Mean CD ↓ | Median CD ↓ | Mean HD ↓ | Pass@1 ↑ | Pass@2 ↑ | Mean CD ↓ | Median CD ↓ | Mean HD ↓ | CF-IoU ↑ |
| *Large-scale Models* | | | | | | | | | | | |
| GPT-4o | **0.836** | **0.932** | 101.431 | 58.381 | 402.292 | **0.811** | **0.904** | 26.940 | 0.902 | 223.191 | 0.133 |
| Gemini-2.5-pro | 0.827 | 0.898 | 97.356 | 51.445 | 380.491 | 0.788 | 0.861 | 20.229 | 0.659 | 179.538 | 0.423 |
| DeepSeek-V3 | 0.832 | 0.926 | 107.509 | 65.556 | 415.350 | 0.795 | 0.892 | 22.718 | 0.713 | 201.850 | 0.383 |
| DeepSeek-R1 | 0.810 | 0.912 | 108.889 | 61.035 | 407.416 | 0.773 | 0.873 | 23.599 | 0.752 | 202.943 | **0.510** |
| *Medium-scale Models* | | | | | | | | | | | |
| Llama-3.1-70B | 0.746 | 0.879 | 113.520 | 74.877 | 443.532 | 0.711 | 0.843 | **23.669** | 1.111 | 219.550 | 0.032 |
| Qwen3-Coder-30B | 0.834 | **0.913** | 103.218 | 66.932 | 423.358 | **0.770** | 0.848 | 23.921 | 0.931 | 210.340 | 0.273 |
| Qwen3-VL-30B | **0.878** | 0.909 | **101.926** | **65.625** | **413.282** | 0.735 | **0.876** | 24.121 | **0.926** | **206.522** | 0.295 |
| *Small-scale Models* | | | | | | | | | | | |
| Qwen3-8B (NoThinking) | 0.688 | 0.808 | 127.014 | 87.839 | 494.014 | 0.639 | 0.755 | 20.549 | **0.786** | **198.497** | 0.210 |
| Qwen3-8B (Thinking) | 0.542 | 0.721 | **107.952** | **73.107** | **433.496** | 0.510 | 0.685 | 21.175 | 0.809 | 201.403 | 0.139 |
| Llama-3.1-8B | 0.691 | 0.849 | 125.984 | 87.964 | 497.800 | 0.624 | 0.778 | 28.924 | 1.312 | 241.715 | 0.044 |
| Qwen3-VL-8B | **0.793** | **0.893** | 127.860 | 87.280 | 479.730 | **0.742** | **0.840** | 19.028 | **0.786** | 200.841 | **0.284** |
| *Supervised Fine-Tuned Models (Ours)* | | | | | | | | | | | |
| Qwen3-8B (FT on Op-CAD) | 0.742 | **0.861** | 117.552 | 75.234 | 441.041 | 0.742 | 0.861 | 24.348 | **0.489** | **154.521** | **0.241** |
| vs. Qwen3-8B | (+7.8%) | (+6.6%) | (-7.4%) | (-14.4%) | (-10.7%) | (+16.1%) | (+14.0%) | (+18.5%) | (-37.8%) | (-22.2%) | (+14.8%) |
| **Op-Llama-8B (Ours)** | **0.746** | 0.835 | **102.247** | **60.890** | **411.358** | **0.783** | **0.887** | 20.633 | 0.856 | 210.089 | 0.120 |
| vs. Llama-3.1-8B | (+8.0%) | (-1.6%) | (-18.9%) | (-30.8%) | (-17.4%) | (+25.5%) | (+14.0%) | (-28.7%) | (-34.8%) | (-13.1%) | (+172.7%) |
| vs. Llama-3.1-70B | (+0.0%) | (-5.0%) | (-9.9%) | (-18.7%) | (-7.3%) | (+10.1%) | (+5.2%) | (-12.8%) | (-23.0%) | (-4.3%) | (+275.0%) |

*Table 6.* Evaluation of DeepCAD and DeepCAD-*Op*. DeepCAD-Op is trained on Op-CAD. Overall, Op-CAD is more challenging than DeepCAD, while training on Op-CAD improves model performance on both benchmarks.

| Model | Test Data | ACC$_{cmd}$ | ACC$_{param}$ | Mean CD | Invalid ratio |
|---|---|---|---|---|---|
| DeepCAD | DeepCAD | 99.28 | 97.46 | 7.10 | 3.80 |
| DeepCAD-*Op* | | 99.32↑0.04% | 97.56↑0.10% | 6.64↓6.48% | 3.43↓9.74% |
| DeepCAD | Op-CAD | 99.14 | 90.65 | 98.54 | 7.30 |
| DeepCAD-*Op* | | 99.52↑0.38% | 96.461↑6.41% | 13.41↓86.39% | 4.95↓32.19% |

(Zhou et al., 2023). Following the pipeline detailed in Section 3, we curated the Op-CAD dataset.

The dataset is split into training, validation, and test sets with a approximate ratio of 8:1:1, where the test set comprises 12,787 modeling tasks. To ensure balanced data distribution and prevent leakage, we perform stratified sampling based on operation types and difficulty levels, while maintaining model-level grouping during the split.

**Baselines.** We conduct a comprehensive evaluation of eight models spanning three scale levels, including closed-source models (e.g., GPT-4o(Achiam et al., 2023) and Gemini-2.5 Pro(Comanici et al., 2025)) and open-source models (e.g., DeepSeek(Guo et al., 2025; Liu et al., 2024) and Llama(Grattafiori et al., 2024)).

### 5.2. Main Results

**Op-CAD Challenges SOTA LLMs.** As shown in Table 5, even state-of-the-art models such as GPT-4o, Gemini-2.5-pro, and DeepSeek-V3 achieve only approximately 80% success on the first attempt (Pass@1) for both tasks, indicating that current models still fall short of reliable task completion. The 3D geometry metrics reveal further limitations. Among large-scale models, Gemini-2.5-pro achieves the best CD and HD on both tasks, while DeepSeek-R1 gains the best CF-IoU at 0.510. However, these values remain relatively high for precise CAD operations, where the values for identical models under the same sample set are approximately 0.3 for CD, 30.7 for HD, and 1.0 for CF-IoU.

**Model Ability varies across Operations.** As shown in Fig. 1, the models exhibit distinct performance patterns across different operation types. Overall, the success rate for extrude operations is higher than that of the other two operations, with chamfer and fillet (CF) showing the lowest success rate among all. This indicates that edge-based operations, such as chamfer and fillet, represent a more challenging spatial reasoning task. Moreover, the 3D metrics reveal that models generally perform worse on revolve operations compared to *extrude*. This disparity can likely be explained by the additional rotation axis parameters introduced in revolve operations, which increase the complexity compared to the linear translation in extrude.

**Inconsistency from Geometry to Spatial Reasoning.** By examining the performance patterns in Table 5, we observe that a model's shape reasoning and spatial reasoning abilities are not necessarily strongly correlated. While medium- and large-scale models exhibit consistent perfor-

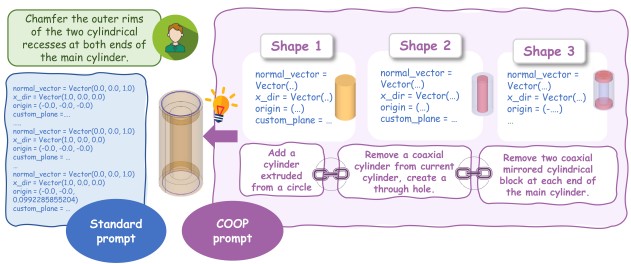

Figure 4. Comparison between the Standard prompt and the COOP prompt. COOP pairs the modeling history with explicit sequence-level descriptions, providing an engineering-style reasoning chain.

mance across both SGM and SSM tasks, a distinct divergence emerges in small-scale models. Specifically, SGM evaluates the understanding of geometric shapes, whereas SSM measures the capacity for spatial positioning and reasoning. This suggests that while large-scale models can bridge the gap between these two capabilities through their vast parameter space, small-scale models—constrained by limited capacity—exhibit a decoupling of these abilities, often succeeding in one while failing in the other. Given the current trend in CAD generation toward unified, end-to-end modeling, our findings suggest that isolating geometric comprehension from spatial reasoning is a promising research direction for developing more precise and efficient CAD-generative AI.

**Effectiveness of the geometry-guided hierarchical annotation pipeline.** We uniformly sample 1,000 data points across four operations to generate sequence-level annotations under three experimental conditions: the complete pipeline, a version without the step-level annotation stage (w/o shape), and a version without geometric guidance (w/o geo). The quality of these annotations is evaluated by presenting images paired with the descriptions generated under the three settings to both GPT-4o and three human experts. Evaluators are tasked with selecting the description that most accurately matches the operation depicted in the image. As shown in Table 8, our complete annotation pipeline achieves significantly higher preference scores than both ablated versions, underscoring the vital importance of hierarchical annotation and geometric guidance in capturing complex CAD operations.

**Models Trained on Code and Engineering Data Perform Better on Op-CAD.** We observe that the Qwen models—pre-trained more extensively on code and engineering-related data—consistently outperform general-purpose models such as the Llama series(Xu et al., 2023). Notably, Qwen3-Coder-30B, which receives substantial code-centric pretraining, surpasses the much larger Llama-3.1-70B and performs on par with DeepSeek-R1. This pattern also appears among smaller models: Qwen-8B out-

Table 7. Results under the COOP strategy for the SSM task. Gains over the Standard setting are highlighted in red, while drops are highlighted in green. Overall, COOP often improves 3D operation accuracy, particularly for common opensource models.

| Model | Pass@1 ↑ | Mean CD ↓ | CF-IoU ↑ |
|---|---|---|---|
| GPT-4o | 0.836↑3.1% | 22.814↓15.3% | 0.108↓18.8% |
| Gemini-2.5-pro | 0.809↑2.7% | 22.852↑13.0% | 0.375↓11.3% |
| DeepSeek-V3 | 0.798↑0.4% | 20.685↓8.9% | 0.291↓24.0% |
| DeepSeek-R1 | 0.788↑1.9% | 21.106↓10.6% | 0.425↓16.7% |
| Llama-3.1-70B | 0.738↑3.8% | 22.941↓3.1% | 0.038↑18.7% |
| Qwen3-Coder-30B | 0.789↑2.5% | 22.334↓6.6% | 0.216↓20.9% |
| Qwen3-8B (NoThinking) | 0.670↑4.9% | 21.024↑2.3% | 0.232↑10.5% |
| Qwen3-8B (Thinking) | 0.533↑4.5% | 19.907↓6.0% | 0.200↑43.9% |
| Llama-3.1-8B | 0.645↑3.4% | 23.834↓17.6% | 0.074↑68.2% |
| **Op-Llama-8B (Ours)** | 0.817↑4.3% | 20.232↓1.9% | 0.163↑35.8% |

Table 8. Ablation studies on the semantic annotation pipeline. *w/o Shape* removes step-level annotation inputs, and *w/o Geo* removes geometric guidance. Our full pipeline achieves the highest preference under both GPT-4o and human evaluations, outperforming the two ablated variants.

| Description Type | GPT-4o Evaluation | Human Evaluation |
|---|---|---|
| Ours | 50.5% | 62.5% |
| Ours *w/o* Shape | 18.5% | 12.5% |
| Ours *w/o* Geo | 31.0% | 25.0% |

performs Llama-3.1-8B across nearly all metrics. The gap is particularly striking for Mean CF-IoU, where Qwen-8B achieves 0.232 compared to only 0.074 from Llama-3.1-8B.

## 5.3. Investigation of Modeling Enhancement Strategies

**Engineering-inspired Chain-of-Operation Prompting (COOP) Enhances Performance.** Inspired by the standard workflow of human engineers—who decompose complex models into geometric structures to determine the precise location and function of a target operation—we propose the Chain-of-Operation Prompting (COOP) strategy. To simulate this human-centric reasoning for the SSM task, we append sequence-level annotations to the preceding modeling scripts. As shown in Table 7, all models exhibit consistent improvements in success rates under the COOP setting. Furthermore, most models show improved CD metrics, and smaller-scale models achieve higher CF-IoU scores compared to the standard setting. In contrast to the observed failure of general Chain-of-Thought (CoT) on the SSM task—where DeepSeek-V3 outperformed R1 and Qwen3-8B performed better in *No-Thinking* mode—COOP represents a promising new direction for exploring effective reasoning paradigms in CAD-related tasks.

**Training on Op-CAD Boosts Model Performance on CAD Tasks.** We supervised fine-tuned Llama-3.1-8B on

the Op-CAD dataset, using the same input and output formats as the SSM task, to develop Op-Llama-8B. As shown in Table 5, Op-Llama-8B significantly outperforms the base Llama-3.1-8B, with 3D metrics improving by an average of 24%, and CF-IoU specifically increasing by 172.7%. Additionally, we train DeepCAD-*op* following the methodology proposed in DeepCAD (Wu et al., 2021). As shown in Table 6, DeepCAD-*op* improves performance on both the original DeepCAD benchmark and Op-CAD. These results collectively demonstrate that the Op-CAD dataset is effective in enhancing the modeling capabilities of models on CAD-related tasks.

# 6. Conclusion

In this paper, we introduce Op-CAD, an operation-oriented dataset that addresses the shortage of complex, model-rich CAD data with diverse operation types. By integrating a CAD parsing module with a hierarchical annotation pipeline, Op-CAD provides fine-grained parsing outputs and structured annotations that capture CAD modeling processes, offering a solid foundation for CAD data construction for future works. Moreover, extensive evaluations with our hierarchical task suite provide valuable insights into current LLMs' geometric and spatial reasoning in CAD. We further fine-tune models on Op-CAD to enhance their CAD modeling capabilities, and propose Chain-of-Operation (COOP) prompting, which encourages human-like CAD reasoning and shows promise as a new paradigm for CAD reasoning. Overall, our work offers important guidance for future research on complex, long-horizon CAD modeling.

## Acknowledgements

This work was supported by the National Natural Science Foundation of China under Grant No. 62506317 and Guangdong Provincial Key Lab of Integrated Communication, Sensing and Computation for Ubiquitous Internet of Things (No.2023B1212010007).

## Impact Statement

This work introduces Op-CAD, a benchmark for language-guided, step-by-step CAD modeling. Our goal is to improve human–CAD interaction by enabling models to assist structured modeling workflows.

**Potential Benefits.** Our approach reduces the learning barrier of CAD systems and improves modeling efficiency by allowing large language models to execute step-by-step modeling instructions. In this framework, the model acts as an execution assistant, while human engineers remain responsible for design intent, verification, and decision-making.

**Potential Risks.** As with many automation technologies, this work may raise concerns about labor displacement in routine CAD tasks. Additional risks include over-reliance on automated outputs, misinterpretation of generated modeling actions, and unequal access to such tools, which may create disparities across users or organizations.

**Data and Ethics.** The dataset is constructed from publicly available CAD modeling sequences released under the MIT License. Our CAD parser module automatically decomposes these sequences into fine-grained operations, without involving private or proprietary data. We will release the dataset, code, and evaluation benchmark to support reproducibility and facilitate future research.

**Mitigation.** We advocate for human-in-the-loop usage, where users retain control and remain responsible for verifying generated designs, especially in safety-critical engineering contexts. We further highlight the need for validation mechanisms, such as geometric consistency checks and constraint-based verification, to improve the correctness and safety of generated CAD models. Improving model reliability, robustness, and transparency remains an important direction for future work.

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

# A. Overview

In the supplementary materials, we provide more detailed technical descriptions of the data annotation pipeline and failure analysis to support the main content of the paper. The contents are organized as follows:

- **Data Annotation Details:** The step-level and sequence-level prompts for VLM.

- **Failure Analysis:** The distribution of failures observed in model testing on Op-CAD, as well as visualizations of representative high- and low-quality generated results.

# B. Data Annotation Details

### B.1. Annotation Prompt Design

In this section, we present the prompt templates used during the data annotation process, which serve as structured inputs to the VLMs.

The **Step-level Prompt**, shown in B.1, is designed for operation shape recognition and is provided to GPT-4o. The inputs include a single sketch image and a three-view rendering of the resulting CAD shape, together with the basic geometric primitives and—when the operation is a revolve—the revolve-axis position information extracted from the geometric parsing module.

The **Sequence-level Prompt**, shown in B.1, is designed for operation position recognition and is provided to Gemini 2.5-Pro. The inputs consist of the acted-object index, its geometric and operation-level descriptions, nine-view renderings of the model before and after the operation, an operation-highlighted image, and textual operation hints. These hints incorporate geometric and spatial cues, including the operation shape description, bounding-box distribution, and acted-surface type.

For each positional label, we define several commonly used linguistic variants—for example, *span_full* corresponds to "penetrating" or "covering"—allowing the model to make visually grounded selections rather than relying on free-form generation. This design substantially reduces hallucination and ensures consistency across annotations.

---

**Step-level Prompt**

**System Prompt**

You are a CAD modeling annotation assistant. Identify the sketch-based shape and provide a concise one-sentence description of the resulting CAD geometry.

**Inputs**

Two images are provided:
1) The **sketch** image.
2) A **three-view** rendering of the resulting CAD shape. Text (from geometric parsing module):
- Basic geometric primitives contained in the sketch.
- Number of repeated shapes and their spatial arrangement.
- (Optional) Other attributes derived from point-pattern analysis (e.g., parallel/perpendicular/filleted relations).

**Task**

- Output **exactly one** short English sentence describing the CAD model *and* its sketch basis.
- Use the form: "A {3D model description} formed by extruding a {sketch description}".
- Avoid phrases like "extrude a line" or "extrude an arc"; describe closed profiles (e.g., arc-bounded, arc-shaped, two pairs of perpendicular parallel lines).

**Constraints**

- Exactly one sentence; no lists, quotes, or extra commentary.
- Keep wording grounded in the provided sketch and three-view images; do not invent geometry.
- If the arrangement is absent or number = 1, omit the arrangement clause.

---

---

**Sequence-level Prompt**

**System Prompt**

You are a CAD modeling annotation assistant. Your goal is to output **exactly one** short English sentence describing a one-step operation on a CAD base shape. Use the phrasing **"Add/Remove ... at/on ..."**, mention the base-shape short name, and never mention step numbers.

**Image Inputs**

You will receive three images in order:
1) One operation-highlighted **mask** image (yellow = added, red = removed, light purple = previous parts).
2) One **BEFORE** image (3×3 multi-view grid; nine views).
3) One **AFTER** image (3×3 multi-view grid; nine views).

**Geometric and Location Information**

**From Geometric Parsing Module**

- Operation type: "{op_type}".
- 3D position labels from bounding box: "{bbox_position}".
- Acted surface label: "{surface_label}".
- Candidate position options (choose, combine, or write your own): "{position_choice}".
- There are "{number}" identical operations arranged as "{arrangement_pattern}". - Independence note (if provided): *This step is independent; no boolean interaction with previous parts.*

**From Shape Annotation Stage**

- Textual shape description of the **operation**: "{op_des}".
- Textual shape description of the **base shape** (the specific part receiving the operation; do not include location): "{acted_des}".

**Task**

• Output **exactly one** short English sentence.
• Use the phrasing **"Add/Remove ... at/on ..."** and include the **base-shape short name** in the location clause.
• Specify *what* (resulting feature; e.g., cylinder, block, cone, boss, rib, slot/pocket, counterbore, countersink, through/blind hole), *where* (precise applied location), and *how many/arrangement* (if multiple identical operations).
• When helpful, use brief modifiers (short/long/flat/thin/thick/small/large) and relative phrasing between base and added/removed features.

**Constraints**

• **Exactly one** sentence; no lists, quotes, prefixes/suffixes, or hedging.
• **Never** mention step numbers (e.g., "step 4"); if you must refer to a previous feature, describe it by geometry or relative position only.
• Only **yellow/red** indicate the current operation in the mask; do **not** treat light purple as part of the added/removed feature.
• Follow the given descriptions of the **operation shape** and the **base shape** strictly; never swap them.

**Incremental Code-Generation Prompt at Test Time.** During testing, we query each LLM with an incremental CadQuery code-generation prompt that conditions on the previously executed code, a natural-language operation description, and the corresponding images. As shown in B.1, the prompt explicitly enforces our shape–then–boolean decomposition and standardizes how new results are linked into the modeling history.

## Incremental CadQuery Code-Generation Prompt

**Role.** *You are an expert CAD modeling assistant specialized in CadQuery. Generate only the incremental CadQuery code needed to perform the requested operation as a continuation of the provided context. The output must be directly executable when appended to the previous code.*

**Inputs.**
- **Previous code context:** the already executed CadQuery script containing variables `result_0`, `result_1`,..., where the last one is treated as the current solid.
- **Operation instruction:** a natural-language description of the next modeling operation to be applied.
- **Optional images:** sketch views, three-view or nine-view renderings, and operation-highlight masks, together with a short image guidance text.

**Linking rules.**
- Treat the latest variable `result_k` as the current solid.
- The new result should be written into `result_k+1` to keep a clear operation history.
- For the very first step, assign the base solid to `result_0`.

**Workplane rules (mandatory).**
- Never use `.faces()` / `.face()` or string shortcuts such as `"XY"`, `"XZ"`, or `"YZ"` to define workplanes.
- Always construct workplanes explicitly with `Plane` and `Vector`, and then create `cq.Workplane(custom_plane)` on top of them.

**Shape–then–Boolean rules (for extrude/revolve/add/cut).**
- In the **#shape** block, the model must build the required feature as an *independent* solid `shape`, without referencing previous `result_k`.
- If multiple bodies are created, they should be united into a single `shape`.
- In the **#bool** block, the model applies exactly one boolean between `result_k` and `shape`, using either `union` or `cut`, and assigns the outcome to `result_k+1`.

**Fillet / chamfer rules (for modification operations).**
- The model must not chain `.fillet()` or `.chamfer()` directly after an edge selector.
- Instead, it first selects edges from `result_k` into a variable (e.g., `edges_1`), and then applies the operation on this selection to produce `result_k+1`.
- This pattern is repeated for subsequent modification steps.

**Hard requirements.**
- Output only the CadQuery code snippet; no natural-language explanation.
- Do not recreate the base model or redefine previous variables unless strictly necessary.
- The code must be syntactically valid and immediately runnable when appended to the context.
- If dimensions are unspecified, choose reasonable proportions relative to local geometry, avoiding self-intersections or infeasible operations.

**Output format (strict).**
- For shape–then–boolean operations:

```
#shape
{generated_shape_code}
#bool
{generated_bool_code}
```

- For fillet / chamfer operations:

```
#edges select
{generated_edges_select_code}
#operation
{generated_operation_code}
```

- Only these structured blocks are allowed; any extra text or alternative formats are considered invalid.

## C. Failure Analysis

**Failure Types and Distribution.** We categorize these failures into two major types. (1) Geometric errors, which include generating invalid topological structures or producing infeasible geometric parameters that cause the CAD operation to fail during execution. (2) Syntactic errors, such as incorrect function calls, malformed arguments, or other code-level mistakes. Notably, geometric errors can be viewed as a higher-level failure mode built upon syntactic correctness—they more directly

reflect the model's actual understanding of geometric constraints and spatial reasoning. Representative error types and examples are shown in Figure 5b. By averaging results across all evaluated models, we obtain the distribution of failure types shown in Figure 5a. The proportion of syntactic errors remains substantially higher, indicating that many models still struggle with basic syntax and therefore fail before their geometric reasoning capabilities can be meaningfully examined.

**Visualization.** We further rendered representative cases to better understand the effects of COOP. Specifically, we selected results with high description–geometry consistency under the COOP setting (Fig. 7) and those with low consistency (Fig. 10), all of which correspond to invalid generations in the Standard setting. These visualizations reveal that COOP enables models to accurately localize features even without precise numerical parameters, relying instead on semantic cues such as "at the center", "adjacent to . . .", or "aligned with . . .". This demonstrates the model's improved ability to interpret relational instructions and place operations at the correct geometric locations.

However, when the underlying CAD model becomes highly complex—especially those involving many similar components or repeated substructures—the models still exhibit notable errors. Typical failure modes include plane mis-rotation, incorrect surface localization, and misaligned or incorrectly arranged repeated features. These observations indicate that, while COOP substantially improves spatial grounding, further refinement is needed for robust localization on complex or highly symmetric models, especially when multiple candidate target regions share similar geometric patterns.

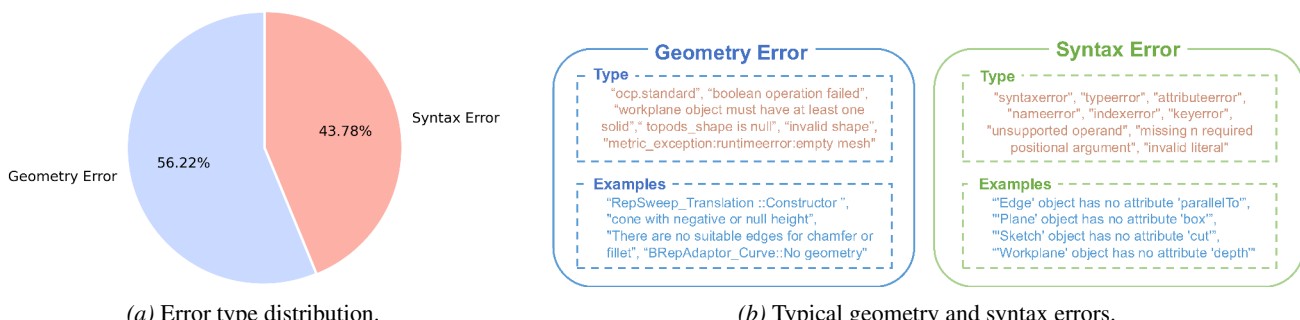

*(a)* Error type distribution.                     *(b)* Typical geometry and syntax errors.

*Figure 5.* Error analysis of CAD code generation.

| single-step shape | Operation | Comparison of Annotation Strategies |
|---|---|---|

Add a circular ring with two rectangular tabs on opposite sides, each containing two circular through-holes, around the thick cylindrical ring at the middle of the syringe body.

Add two flanges with circular holes symmetrically on the exterior of the central circular body.

Add two rectangular mounting flanges with through holes on the side surface of the circular housing, positioned opposite each other.

Remove a disk with a central pentagonal through-hole from the center of the large cylindrical step of the main body, creating an external pentagonal step.

Remove a cylindrical through hole coaxial with the main axis of the long cylinder.

Remove a disc-shaped solid with a hexagonal through-hole on the large square flange.

Add a rectangular prism at one end of the top surface of the block with a recessed section.

Add a rectangular block on the top face of the back wall of the U-shaped plate.

Add a rectangular block on the rear section of the C-shaped base.

Chamfer on the four outer top edges of the square plate.

Chamfer all the edges of the rectangular base.

Chamfer at the four outer top edges of the rectangular base and the two vertical edges of the extrusion.

*Figure 6.* Comparison of annotation generated by the full three-stage annotation pipeline and its two ablated variants (Ours, Ours w/o Step, Ours w/o Geo)

# ✔️ High-quality Results

| Instruction | Ground Truth | Generated model |
|---|---|---|

The orange markings indicate important positional cues.

Remove a hexagonal pocket **at the center of the flat end** of the chamfered cylinder.

Add a flared cylindrical boss on the t op face of the rectangular block, **creating a uniform line of three bosses along one long edge**.

Remove a through hole **coaxial** with the base axis **at one end of the hexagonal prism**.

Add a short, wide cylinder at the **center of an end face of the tall,** narrow cylinder.

Add two mirrored short cylinders **at the end cap** of the tall rectangular prism, centered **along its thick dimension** and **positioned towards its short edges**.

Remove a capsule-shaped through-slot at the **top face of the rectangular prism**, **near the corner** opposite to the first capsule-shaped through-slot.

17

*Figure 7.* Examples of generated high-quality results.

# ❌ Low-quality Results

| Instruction | Ground Truth | Generated model |
|---|---|---|

The orange markings indicate important positional cues.

| | | |
|---|---|---|
| Remove two coaxial chamfered ring grooves, **vertically stacked**, at the edges of the top and bottom end caps of the hollow hexagonal prism. | | |
| Remove a hexagonal recess at the center of the end cap of the **larger cylinder**. | | |
| Add a three-quarter cylinder at the center of the ring's **outer part**, along its **outer surface**. | | |
| Add a pair of hexagonal bosses **on the both side of the small rectangular protrusion,** arranged symmetrically as mirror counterparts. | | |
| Add a short cylinder to the end cap of the rounded octagonal boss that sits on the side of the main body. | | |
| Remove the four short cylinders from the side surface of the thin rectangular block on the side of the main body, which are aligned in a row along the edge near the rectangular recess. | | |

18

*Figure 8.* Examples of generated low-quality results.

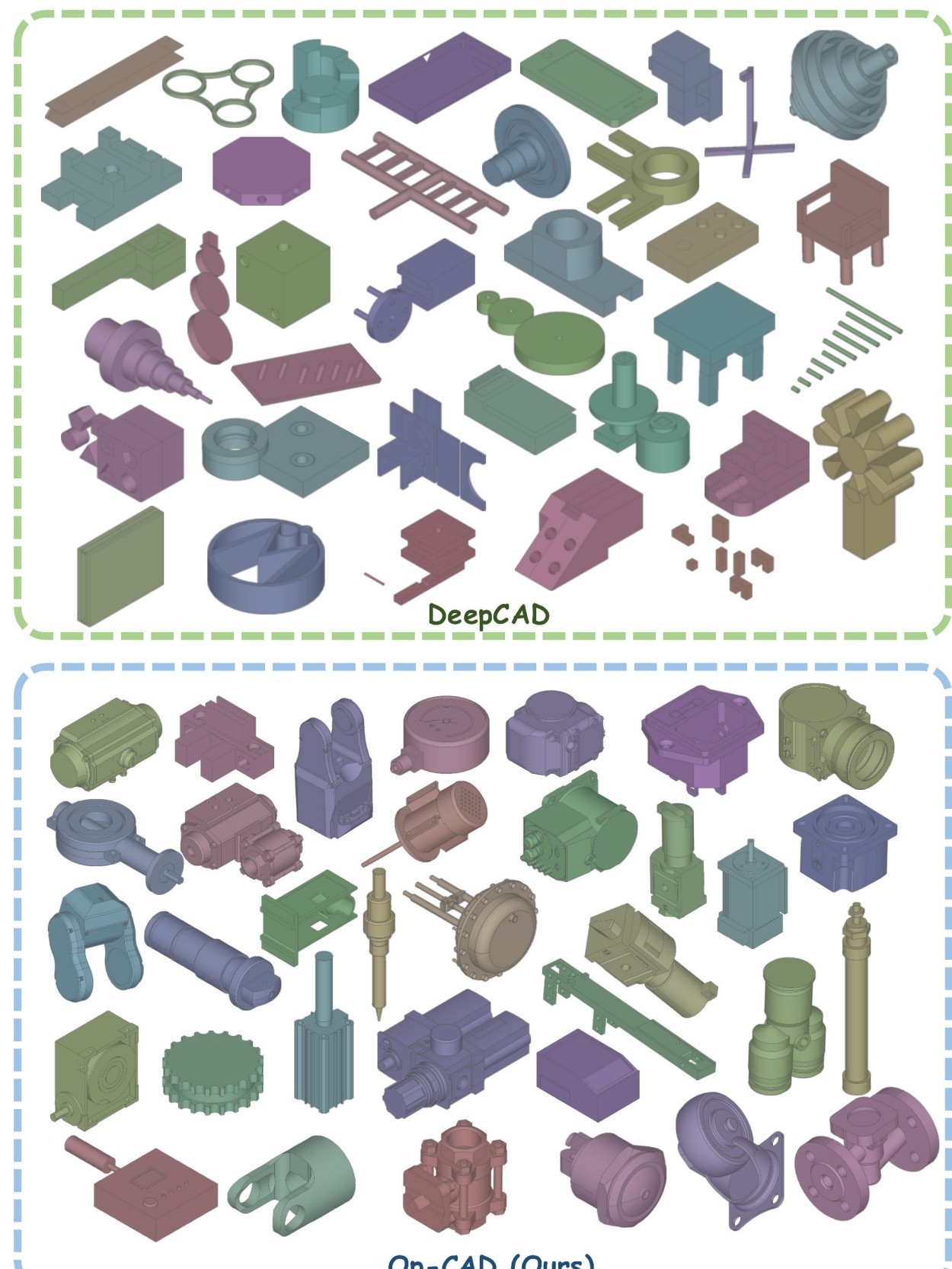

*Figure 9.* Figure 9. Comparison between DeepCAD and our Op-CAD models. Samples are selected by descending operation sequence length from valid reconstructions. Our Op-CAD dataset exhibits higher complexity and more meaningful geometric structures.

## Sequence Modeling Visualization in Op-CAD

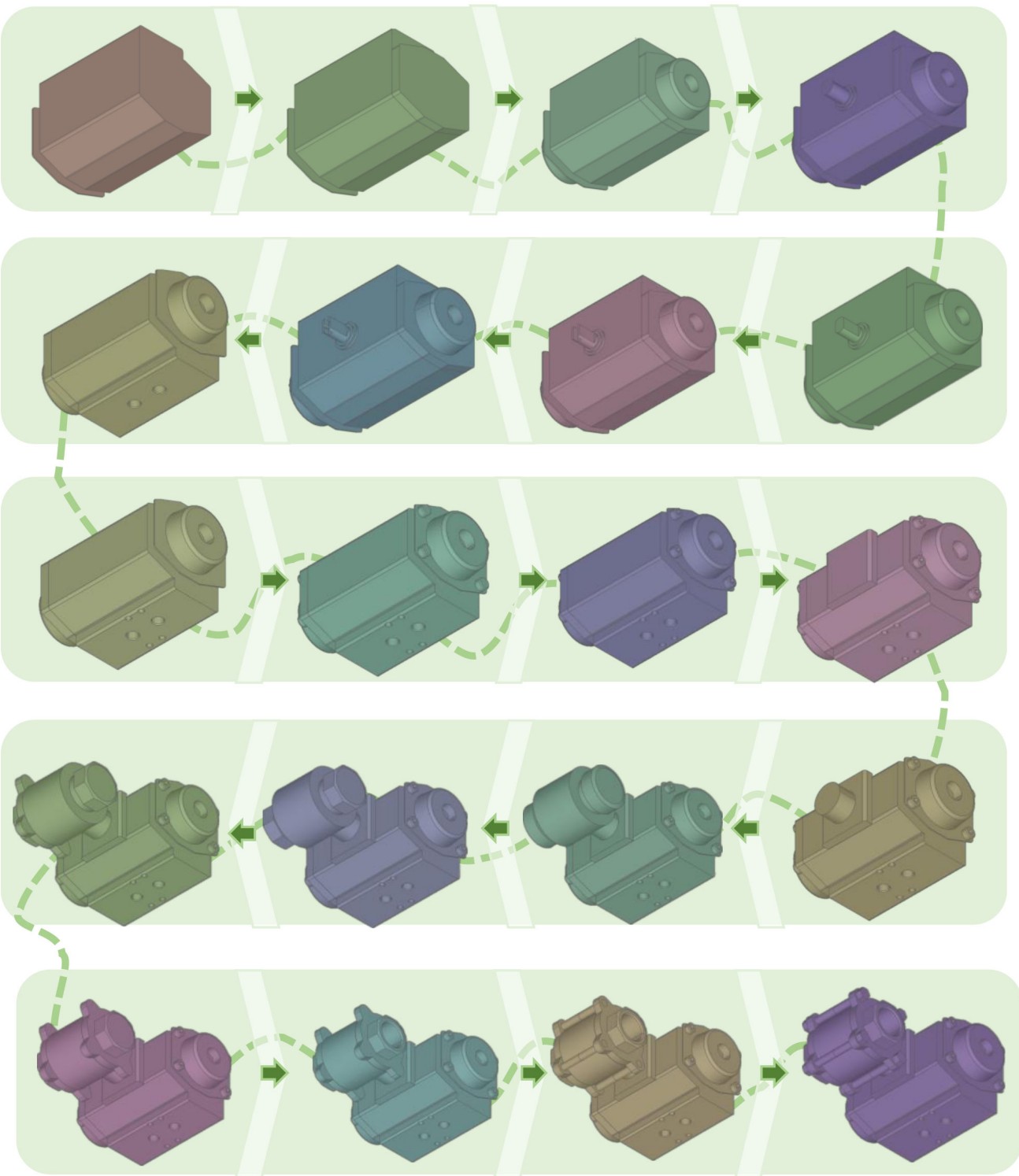

*Figure 10.* A visualization example of sequence-based modeling in Op-CAD, illustrating how a CAD model is progressively constructed through an ordered sequence of operations. Each step reflects a specific geometric transformation, highlighting the intermediate states and the evolution of the model toward the final structure.

