# OpenReview forum: "Op-CAD: Benchmarking and Investigating Operation-oriented CAD Generation"
_ICML.cc/2026/Conference — ICML 2026 regular_

### Official Review · Reviewer_cMnX · 2026-02-26

**Soundness:** 3
**Presentation:** 2
**Significance:** 3
**Originality:** 2
**Overall Recommendation:** 4
**Confidence:** 4

**Summary:**

This paper presents Op-CAD, a large-scale benchmark for operation-oriented CAD generation with 128k operation instances spanning multiple operation types and modalities. It proposes a hierarchical evaluation protocol and introduces CF-IoU to measure advanced operations such as chamfer/fillet. The paper evaluates a range of LLM/VLM-based methods, studies prompting strategies (e.g., COOP prompting), and shows that fine-tuning on Op-CAD (e.g., Op-Llama-8B) yields substantial gains on geometric and operation-specific metrics.

**Compliance With Llm Reviewing Policy:**

Affirmed.

**Final Justification:**

Thanks for the additional clarification. The follow-up addresses my earlier concerns more directly, especially regarding the interpretation of CF-IoU and the robustness of the benchmark under annotation noise. I appreciate these additions. That said, while my confidence in the paper has improved, I still view the overall submission as falling short of the weak-accept threshold, so I am not changing my score.

**Key Questions For Authors:**

1. Please provide a precise definition of CF-IoU and analyze its behavior on controlled cases (varying fillet radius, chamfer size, model scale).
2. What is the label/QC pipeline and error rate? How do labeling errors affect evaluation conclusions?
3. How does Op-CAD compare (coverage and difficulty) to the closest existing CAD datasets/benchmarks?
4. What is the plan for dataset release and licensing? If partial release, how can the community reproduce results?

**Limitations:**

Not fully. Please add a dedicated section on metric limitations (CF-IoU failure cases), dataset bias (software/format dependence), and licensing constraints.

**Strengths And Weaknesses:**

Strengths
1. Strong dataset/benchmark contribution: operation-level CAD generation is under-evaluated, and long-sequence procedural modeling is practically relevant.
2. Introduces operation-aware metrics (CF-IoU) aiming to cover gaps in traditional geometric similarity metrics.
3. Provides systematic model comparisons and shows tangible fine-tuning benefits.

Weaknesses
1. CF-IoU and the hierarchical evaluation protocol need more formal definition and property analysis (sensitivity to scale/curvature, robustness to topology changes).
2. Data construction and labeling QC need clearer documentation (noise sources, error rates, manual verification protocol).
3. Licensing/redistribution of CAD assets may be non-trivial; the paper should clarify legal/ethical compliance and release plan.
4. Impact statement appears insufficiently detailed (should be treated seriously).

---

> ### Author Rebuttal · Authors · 2026-03-31
>
> **We sincerely thank the reviewer for the constructive and thoughtful feedback.**
> ***
> ### 1. Formal Definition and Properties of CF-IoU
> *(Response to “Weakness 1” and “Question 1”)*
>
> We thank the reviewer for the insightful comments on the CF-IoU metric and the hierarchical evaluation protocol.
>
> **(1) Formal definition.**
> Let $E_{gt}$ denote the set of ground-truth edges to be modified, and $E_{pred}$ denote the predicted set of edges extracted from the generated CAD program via standard CAD APIs. CF-IoU is defined as:
> $$\mathrm{CF\text{-}IoU} = \frac{|E_{gt} \cap E_{pred}|}{|E_{gt} \cup E_{pred}|}$$
> We note that a potential source of confusion may be what is captured in the edge set $E$. Here, $E$ refers to the **indices of target edges** to which the operation should be applied, rather than including operation parameters such as chamfer width or fillet radius. This design reflects the nature of CAD modeling: the primary challenge of chamfer/fillet operations lies in **correct edge localization**, while parameter specification is comparatively straightforward once the target edges are identified.
>
> **(2) Invariance properties.**
> CF-IoU is inherently **invariant to scale and geometric parameters** (e.g., fillet radius, chamfer size, and model scale), as its definition does not include operation parameters.
> Moreover, as long as the target edges are selected, they can be consistently identified through CAD APIs (e.g., CadQuery). Therefore, CF-IoU is robust to geometric variations and transformations. For topology changes that preserve the underlying target edges, CF-IoU remains unaffected.
>
> We will include a clearer formal definition and discussion of these properties in the revised paper.
> ***
> ### 2. Data Construction and Quality Control
> *(Response to “Weakness 2” and “Question 2”)*
>
> We thank the reviewer for emphasizing the importance of data quality and documentation.
>
> - **Noise sources and error characteristics.**
> The primary noise source comes from VLM-generated natural language annotations, which may introduce ambiguity or hallucinations. Due to the open-ended nature of language, a strict error rate is difficult to define. However, ablation results (Table 8) show that incorporating geometric and structural information significantly improves annotation quality, as reflected by higher winning rates.
>
> - **Manual verification protocol.**
> We incorporate human verification for complex cases. In particular, for revolve operations where sketches and shapes are more intricate, we apply automatic filtering to remove simple cases and manually correct the remaining annotations. This process covers approximately **19%** of sketches.
>
> - **QC pipeline.**
> After annotation, we perform consistency checks using our CAD parsing module. Specifically, we leverage structured information parsed from our parsing module to detect and correct inconsistencies, ensuring that the final annotations are **factually consistent** with the underlying CAD sequences.
>
> We will further document these processes in the revised paper to improve clarity and reproducibility.
> ***
> ### 3. Comparison with Existing CAD Datasets
> *(Response to “Question 3”)*
>
> We thank the reviewer for this suggestion.
>
> Most recent open-source CAD datasets (e.g., Text2CAD, CAD-Editor, GenCAD-Code) are built upon or derived from DeepCAD; therefore, the comparison with DeepCAD in Table 4 is representative. Following the reviewer’s suggestion, we further compare Op-CAD with a more recent dataset, Omni-CAD, in terms of sequence complexity:
>
> | Dataset  | Avg Length | Max Length | [0–3] | [4–20] | [21–50] | [51–105] | [106–277] |
> |----------|-----------|------------|-------|--------|---------|----------|-----------|
> | **Op-CAD**   | **14.29** | **277** | 32.42% | 50.48% | 9.50%  | 5.86%   | 1.74% |
> | Omni-CAD | 2.24  | 102 | 85.67% | 14.06% | 0.21%  | 0.06%   | 0.00% |
>
> From this comparison, Op-CAD exhibits **substantially longer sequences and a much higher proportion of complex, long-horizon samples**, indicating significantly increased modeling difficulty.
>
> In addition, Omni-CAD and DeepCAD primarily focus on extrusion operation (Table 1), whereas Op-CAD introduces multiple operations (Extrude, Revolve, Fillet, Chamfer), providing **both greater sequence depth and richer operational diversity**.
> ***
> ### 4. Dataset Release and Licensing
> *(Response to “Weakness 3” and “Question 4”)*
>
> We will release Op-CAD upon acceptance, including data, annotations, and code (CC BY 4.0 + MIT). Full pipelines and configurations will be provided for reproducibility, with legal compliance clarified in the revision.
> ***
> ### 5. Impact statement
> *(Response to “Weakness 4” )*
>
> We thank the reviewer for this important suggestion. We will expand the Impact Statement in the revision, including a balanced discussion of positive impacts, potential risks (e.g., IP, safety, and employment), and corresponding mitigation strategies (e.g., open-source release and ethical guidelines).

---

> > ### Author Rebuttal · Reviewer_cMnX · 2026-04-01
> >
> > Thanks for the rebuttal. The response clarifies several points, especially the intended definition of CF-IoU and the additional comparison to Omni-CAD, which I found helpful. However, my concerns are only partially resolved. In particular, I still think the paper would benefit from a clearer discussion of what CF-IoU captures versus ignores, as well as a more quantitative characterization of annotation/QC errors and their potential effect on the benchmark conclusions. Overall, the rebuttal improves the paper’s clarity, but not enough for me to change my score.

---

> > > ### Author Response · Authors · 2026-04-02
> > >
> > > **We thank the reviewer for the insightful and constructive suggestions, which helped us better clarify our work. Regarding your follow-up questions, our responses are as follows.**
> > > ***
> > >
> > > ### Further Clarification about CF-IoU
> > >
> > > All factors that influence the quality of chamfer/fillet operations can be explicitly decomposed into three components:
> > >
> > > (1) **Edge selection**: which edges the operation is applied to
> > >
> > > (2) **Operation feasibility**: whether the operation can be successfully executed
> > >
> > > (3) **Parameter specification**: e.g., fillet radius or chamfer size
> > >
> > > - CF-IoU **captures** (1), i.e., whether the model selects the correct target edges, as (1) is the most challenging part of chamfer and fillet operations and has the highest error rates.
> > >
> > > In our evaluation,  **(2) operation feasibility** can be evaluated by Pass@k. Regarding **(3) parameter specification**,  we observe that all models exhibit strong instruction-following ability and achieved over 99% (mean = 99.3%) on parameter accuracy. Therefore, (3) does not significantly affect the evaluation results.
> > >
> > > ---
> > >
> > > ### Quantitative Characterization of the Annotation Pipeline and Quality Control
> > >
> > > To further quantify annotation composition, we report the proportion of verifiable content versus non-verifiable VLM-generated content using our QC strategy:
> > >
> > > | Annotation Stage | Verifiable (Count) | Verifiable (Token Ratio) | Non-verifiable (Count) | Non-verifiable (Token Ratio) |
> > > |------------------|------------------|-------------------------|------------------------|------------------------------|
> > > | **Step-level**   | 75.93% (89,924) | 68.24% (1,846,062) | 24.07% (28,506) | 31.76% (859,279) |
> > > | **Sequence-level** | - | 17.41% (3,175,726) | - | 82.59% (2,622,763) |
> > >
> > >
> > >
> > > ### Impact of Annotation Noise on Benchmark
> > >
> > > To quantify the effect of potential errors introduced by unverified VLM-generated content, we construct a **high-noise subset (Q4)** by ranking samples according to the proportion of VLM-generated tokens in their annotations and selecting the top quartile.
> > >
> > > We then compare model performance rankings between the full test set (**ALL**) and this high-noise subset (**Q4**).
> > >
> > > As shown below, we observe consistently high rank correlation (**Spearman ρ ≈ 0.85–0.96**) and strong pairwise consistency (**≈0.84–0.96**) across key metrics, demonstrating that benchmark conclusions are robust to potential errors in VLM-generated annotations.
> > >
> > > | Metric | Spearman ρ (ALL vs Q4) | Pairwise Consistency |
> > > |--------|------------------------|----------------------|
> > > | **SGM Pass@1** | 0.85 | 0.846 |
> > > | **SGM Mean CD** | 0.84 | 0.844 |
> > > | **SGM Mean HD** | 0.91 | 0.913 |
> > > | **SSM Pass@1** | 0.87 | 0.869 |
> > > | **SSM Mean CD** | 0.91 | 0.906 |
> > > | **SSM Mean HD** | 0.96 | 0.957 |
> > > | **SSM CF-IoU** | 0.95 | 0.945 |

---

### Official Review · Reviewer_YyN2 · 2026-03-04

**Soundness:** 2
**Presentation:** 2
**Significance:** 2
**Originality:** 3
**Overall Recommendation:** 3
**Confidence:** 2

**Summary:**

This paper is about CAD generation (cadquery), mainly a benchmark with 128k operation-level instances. Its inherently a multimodal task, breaked down into 2 levels: first step-level, then sequence-level. The authors claim that this can decouple geometric and spatial evaluation. Tested on Llama-3.1-8B, Qwen3-Coder-30B, Qwen3-8B, etc.

**Compliance With Llm Reviewing Policy:**

Affirmed.

**Final Justification:**

Thank you. As I mentioned previously:

> My biggest concern is the contribution of this work to the AI community. First, the method is fundamentally zero-shot evaluation and SFT-only training. The coop is prompt engineering...

The authors state:

> we provide a high-quality dataset for the data-scarce CAD generation domain, and demonstrate its training value through both SFT (Table 5) and DeepCAD-style training (Table 6). Building on this, we introduce a task-specific reasoning paradigm (COOP) grounded in CAD operation sequences, supported by consistent empirical improvements across models (Table 7).

So I lean towards my initial judgment.

**Key Questions For Authors:**

- Authors emphasize their dataset is more complex than deepcad. As I understand, this is a big reason why authors introduced **step-level** and sequence-level procedures. But in the examples, I didnt see any quite complex examples on page 16. Also, page 16 and page 17 doesnt seem very pleasing to eyes.

- Authors say "comprehensive ... evaluation" on the right of line 72, but from table 1 the operation only covers up to 4 types.

- Please explain why the CF metric is generalizable for the cad domain, instead of just cherry-picking for your constructed dataset. As I understand, this definition relies on "chamfer/fillet". Are these kinds of metrics effective for many other "local features", like hole, shell thickness etc?

- some wording is entangled with ML-field wording. Line 60 right: sequence length vs token sequence length.

- some notation, like the *Sketch* in Line 298 and right after in Line 299 is not very pleasing to eyes.

- Is it proper to call GPT, Gemini etc large scale models? Or proprietary models?

**Limitations:**

yes

**Strengths And Weaknesses:**

Strengths:
- Dataset scale is good.
- sequence length and diversity is better than DeepCAD,
- introduced new evaluation metrics, e.g., CF-IoU.

Weaknesses:
- My biggest concern is the contribution of this work to the AI community. First, the method is fundamentally **zero-shot evaluation** and **SFT-only training**. The coop is **prompt engineering**, and, my concern is, not much real innovation.
- Second, about the dataset, authors also point out DeepCAD dataset already exists, but this new proposed dataset just increases *sequence depth and operational diversity*. For the *sequence depth*, however, in the examples, I didnt see any high-sequence-depth examples on page 16. For the *operational diversity*, seems only 4 in Table 1. And I doubt whether 2 out of the 4, *Chamfer* and *Fillet*, effective for many other "local features" in other CAD data, like hole, shell thickness, etc.

---

> ### Author Rebuttal · Authors · 2026-03-31
>
> **We sincerely thank the reviewer for recognizing the scale of our dataset and the novelty of our metrics, as well as for the thoughtful and responsible feedback, which is very valuable for improving our work.**
> ***
> ### 1. Contribution Beyond Prompt Engineering
> *(Response to “Weakness 1”)*
>
> We thank the reviewer for this important concern.
>
> We refer the reviewer to our response to **Reviewer ExkR (Weakness 1 & 2)**, where we provide a detailed discussion on the novelty and significance of our contributions.
>
> In brief, we provide a high-quality dataset for the data-scarce CAD generation domain, and demonstrate its training value through both SFT (Table 5) and DeepCAD-style training (Table 6). Building on this, we introduce a task-specific reasoning paradigm (COOP) grounded in CAD operation sequences, supported by consistent empirical improvements across models (Table 7).
> ***
> ### 2. Dataset Complexity and Coverage
> *(Response to data complexity mentioned in “Weakness 2” and “Question 1”)*
>
> We thank the reviewer for the valuable feedback on dataset complexity and presentation.
>
> **(1) Sequence depth and modeling complexity.**
> Prior works have consistently shown that **sequence length and operation complexity are key factors limiting CAD modeling performance**. In particular, HNC-CAD[1] (Sections 1–2) shows that modeling difficulty increases **exponentially** with longer sequences and more complex operation dependencies. Op-CAD exhibits the longest sequence lengths among publicly available datasets and introduces additional operations compared to DeepCAD, substantially increasing modeling difficulty.
>
> **(2) Operation coverage and diversity.**
> While Op-CAD includes four core operations (Extrude, Revolve, Fillet, Chamfer), these are in fact the fundamental building blocks of industrial CAD modeling:
> - **Most commonly used operations.**  Tutorials from professional CAD systems (e.g., SolidWorks, Fusion 360) show that 70–80% of modeling workflows rely on these four operations.
> - **Expressive completeness.**  Moreover, many other operations (e.g., Hole, Shell) can be directly constructed using combinations of Extrude and Revolve, indicating that this operation set is sufficiently expressive.
> ***
> ### 3. Complex Model Visualization and Presentation Improvement
> *(Response to "Question 1" on example complexity and visualization)*
>
> We thank the reviewer for the valuable feedback on visualization.
>
> We make two improvements accordingly:
>
> - **Comparison of complex models.**
> We provide additional visualizations comparing complex models from Op-CAD and DeepCAD (Fig. 9), as well as a step-by-step construction example from Op-CAD (Fig. 10), to better illustrate sequence complexity.
>
> - **Improved figure quality.**
> We refine the visualization in Fig. 7 and Fig. 8 (Pages 16–17) to improve clarity and presentation quality.
>
> Updated results are provided at the anonymous link below:
> https://anonymous.4open.science/r/mode-visualization/icml2026figure.pdf
> ***
> ### 4. Generalizability of CF-IoU
> *(Response to “Question 3”)*
>
> We thank the reviewer for this important question.
>
> - **Independence from dataset.**
> CF-IoU is not tied to our dataset, but is defined based on **CAD operations (Chamfer/Fillet)** parsed from standard CAD APIs (e.g., CadQuery). Therefore, it is applicable to any CAD model containing these operations, independent of specific data sources.
>
> - **Operation-specific necessity.**
> There is no need to use CF-IoU for other local features such as Hole or Shell, as they can be directly evaluated using standard 3D metrics (e.g., Chamfer Distance and Hausdorff Distance). In contrast, CF-IoU is specifically designed for **edge-based modification operations** (chamfer/fillet), whose effects are not reliably captured by point-cloud-based metrics. Therefore, CF-IoU provides a necessary complement for accurately evaluating these operations.
> ***
> ### 5. Wording, Notation, and Terminology
> *(Response to "Question 4", "Question 5" and "Question 6")*
>
> We thank the reviewer for these helpful suggestions on wording and presentation.
>
> - **Sequence length terminology:**   We agree that the current wording may be confusing. We will revise it to **operation sequence length** to clearly distinguish it from token sequence length.
> - **Notation consistency:**  We will improve notation consistency by standardizing CAD-specific terms (e.g., *Sketch*, *Revolve*) using consistent italic formatting.
> - **Model categorization:**  We agree with the reviewer’s suggestion and will revise the terminology to more appropriately distinguish these models (e.g., referring to GPT/Gemini as proprietary models).
>
> [1] Xu, X., Jayaraman, P. K., Lambourne, J. G., Willis,K. D., and Furukawa, Y. Hierarchical neural coding
> for controllable cad model generation. arXiv preprint, arXiv:2307.00149, 2023.

---

> > ### Author Rebuttal · Reviewer_YyN2 · 2026-04-02
> >
> > Thank you for the detailed rebuttal.
> >
> > The additional figures and explanations regarding the dataset are helpful and have addressed my concerns on dataset clarity.
> >
> > However, regarding the core contribution, I still feel that the overall novelty remains largely engineering-driven.
> >
> >  I appreciate the authors’ efforts in improving the paper.

---

> > > ### Author Response · Authors · 2026-04-03
> > >
> > > ### Your remaining concern is not a weakness but a feature.
> > >
> > > We respectfully hope the reviewer will reconsider the positioning of this work. Rather than pure methodological innovation, we introduce a new problem setting and paradigm by identifying a critical gap between LLMs and operation-oriented spatial reasoning and modeling in CAD. Building on this core gap, our work contributes a complete and well-founded framework.
> > >
> > > Beyond engineering-driven improvements, our work introduces substantial and novel contributions—also **recognized by Reviewer tWq9**—including:
> > >
> > > * **Dataset & construction:** A pipeline that injects structured geometric knowledge into CAD sequences, enabling step-level supervision, along with Op-CAD—the first operation-level dataset with rich operations and high complexity.
> > > * **Evaluation:** A novel hierarchical protocol for fine-grained, step-wise and sequence-level alignment, with benchmarking that reveals limitations of state-of-the-art LLMs.
> > > * **Methods:** Both training-free and training-based approaches, including operation-oriented SFT and a human-engineering-inspired diagram-COOP strategy, achieving significant improvements.
> > >
> > > Also, we respectfully note that **engineering-driven is not a weakness but a feature**. AI for CAD or relevant engineering fields requires intensive domain engineering knowledge and skills. We argue that this should not be a major weakness for rejecting a paper in this field, if there is no other hiden concern. Incorporating the engineering skills and knowledge into AI for CAD can be a significant contribution to our community.

---

### Official Review · Reviewer_ExkR · 2026-03-13

**Soundness:** 3
**Presentation:** 2
**Significance:** 3
**Originality:** 2
**Overall Recommendation:** 4
**Confidence:** 3

**Summary:**

The authors attempt to address a general context of evaluating LLMs for CAD modeling. This article's primary topic comprises the introduction of Op-CAD, a dataset and benchmark for operation-oriented CAD generation, which decomposes CAD modeling into step-level geometric generation and sequence-level spatial reasoning. The paper also proposes a parsing and annotation pipeline, a new metric (CF-IoU) for chamfer/fillet evaluation, and studies prompting and fine-tuning strategies on the dataset.

**Compliance With Llm Reviewing Policy:**

Affirmed.

**Final Justification:**

After reading the authors' rebuttal, I agree that the lack of a dataset is a bottleneck in CAD. The proposed dataset addresses this issue and I tend to accept the paper.

**Key Questions For Authors:**

How well would models trained on Op-CAD generalize to real industrial CAD systems or different CAD formats?

**Limitations:**

N.A.

**Strengths And Weaknesses:**

++
1. The paper introduces a relatively large and multimodal CAD dataset, which is valuable given the lack of high-quality data in this area.
2. The formulation that separates geometric reasoning and spatial reasoning is interesting and helps better analyze model behavior.
3. The experiments cover multiple LLMs and provide useful observations, e.g., performance gaps across operation types and between geometry vs. spatial reasoning.

-
1. The main contribution is mostly dataset and benchmarking; methodological novelty is limited.
2. Some components (e.g., annotation pipeline, prompting strategy) feel like engineering combinations of existing ideas rather than fundamentally new contributions.
3. It is still unclear how well this benchmark reflects real industrial CAD workflows (e.g., constraints, iterative editing).
4. The evaluation is mainly metric-driven; more discussion on practical usefulness or failure cases would strengthen the paper.

---

> ### Author Rebuttal · Authors · 2026-03-31
>
> **We sincerelythank the reviewer for the insightful and constructive feedback, as well as the recognition of our work.**
> ***
> ### 1. Novelty of Contributions
> *(Response to “Weakness 1” and “Weakness 2”)*
>
> We would like to clarify that high-quality datasets remain a key bottleneck in CAD generation. In this context, large-scale datasets and benchmarks constitute **core contributions** that enable meaningful progress in the field. Indeed, several recent impactful works (e.g., Large Language-Geometry Model, MV-MATH, OmniBench) are centered around dataset and benchmark design.
>
> Beyond the dataset itself, our work introduces **task-driven innovations that enhance model understanding in the CAD domain**:
>
> - **New task formulation:** We are the first to formulate CAD modeling as a  language-guided, step-by-step sequential process , enabling long-horizon spatial reasoning. Unlike the dominant one-shot paradigm, this formulation captures the procedural nature of CAD modeling and allows models to reason over the construction process.
>
> - **COOP prompting strategy:**  COOP is a task-specific Chain-of-Operation strategy for sequential CAD modeling, rather than generic step-by-step prompting. As shown in Table 5, generic reasoning (e.g., Qwen3-thinking, DeepSeek-R1) does not improve performance, highlighting the need for operation-aware guidance. COOP provides such structure and consistently improves performance (Table 7).
>
> - **Hierarchical annotation pipeline with CAD parsing module:** We are the first to integrate VLMs with CAD geometric and topological contexts, effectively mitigating hallucinations. Our ablation results (Table 8) confirm that this full design is essential, clearly indicating that our approach goes beyond trivial engineering. To further solidify this claim, the comparison with prior methods below demonstrates that our pipeline generates richer, more fine-grained multimodal annotations while requiring minimal input.
>
> | Method      | Pipeline Input                     | Pipeline Structure                                   | End-to-End | Modalities Generated           | Annotation Level            |
> |-------------|----------------------------------|------------------------------------------------------|-----------|-------------------------------|-----------------------------|
> | Text2CAD    | Multi-view images + command seq. | Semantic annotation                                  | ✗         | 1 (text)                      | Shape-oriented              |
> | CADmium     | Multi-view images + command seq. | Semantic annotation                                  | ✗         | 1 (text)                      | Shape-oriented              |
> | CAD-LLaMA   | Component images                 | Hierarchical semantic annotation                     | ✗         | 1 (text)                      | Shape-oriented              |
> | **Op-CAD (ours)** | **Original CAD sequence**        | **Parsing module + hierarchical annotation**         | ✓         | **4 (text, image, step, code)** | **Operation-oriented**      |
> ***
> ### 2. Industrial Relevance and Generalization
> *(Response to “Weakness 3” and “Question 1”)*
>
> We thank the reviewer for raising this important point.
>
> We acknowledge that Op-CAD does not yet cover all aspects of real industrial CAD workflows (e.g., constraints and iterative editing). However, we would like to emphasize that Op-CAD is, to the best of our knowledge, **the closest benchmark to real-world mechanical CAD workflows among existing Text-to-CAD datasets**, for the following reasons:
>
> - **Real-world data source:** All data are derived from professional mechanical modeling software(SolidWorks), not synthetic geometry.
> - **Long-horizon modeling sequences:**  Avg. 14.29 steps, better reflecting the multi-step, dependency-heavy nature of industrial CAD design (see Tables 3–4).
> - **Task decomposition aligned with CAD workflows:** We explicitly model **Localization** and **Interaction** (section 4.1), which correspond to key stages in real CAD workflows (e.g., sketching → positioning → boolean/refinement operations).
>
> Regarding **generalization**, we observe that models trained on Op-CAD also achieve improvements on external benchmarks such as DeepCAD (Table 6), suggesting that the learned **operation-level reasoning** exhibits a degree of cross-dataset transferability.
> ***
> ### 3. Practical Usefulness and Failure Analysis
> *(Response to “Weakness 4”)*
>
> We already provide a failure analysis in Appendix C (including error distribution and representative cases), as well as additional failure examples in Fig. 8, illustrating common issues such as incorrect spatial localization, repeated structures, and erroneous topological interactions.
>
> We thank the reviewer for this suggestion. As suggested, we will further expand the discussion on practical usefulness and provide more in-depth failure analysis in the revised paper.

---

### Official Review · Reviewer_tWq9 · 2026-03-13

**Soundness:** 3
**Presentation:** 4
**Significance:** 3
**Originality:** 3
**Overall Recommendation:** 4
**Confidence:** 4

**Summary:**

This paper introduces Op-CAD, a large-scale multimodal dataset and evaluation framework for studying operation-oriented CAD modeling with LLMs. The authors aim to address the lack of datasets that capture step-by-step CAD modeling processes and provide fine-grained supervision for CAD generation.

Based on Op-CAD, the paper further proposes a hierarchical evaluation framework that separates two tasks: Step-level Geometric Modeling (SGM), which evaluates the generation of individual CAD operations. Sequence-level Spatial Modeling (SSM), which evaluates reasoning over the full modeling sequence.

The paper also introduces a new metric, CF-IoU, designed to evaluate chamfer and fillet operations. The authors benchmark several LLMs and propose a prompting strategy called Chain-of-Operation Prompting (COOP) to improve spatial reasoning during CAD generation.

**Compliance With Llm Reviewing Policy:**

Affirmed.

**Final Justification:**

Thank you for your response. While there is still room for improvement regarding the dataset, the authors have addressed most of my concerns. Therefore, I would like to maintain my original score.

**Key Questions For Authors:**

1. The proposed COOP strategy is mainly evaluated on the SSM task. It would be helpful to evaluate COOP on additional CAD-related tasks. Such experiments would help clarify whether COOP provides general improvements for CAD reasoning or primarily benefits the specific task formulation introduced in this work.

2. The experiments mainly compare general-purpose LLMs. Comparisons with domain-specific CAD models would provide a more comprehensive evaluation of the Op-CAD benchmark.

3. Most results focus on prompt-based evaluation. Since Op-CAD is a large-scale dataset with rich annotations, it would also be valuable to demonstrate its usefulness for model training or fine-tuning.

**Limitations:**

yes

**Strengths And Weaknesses:**

Strength:
1. The paper presents a clearly defined pipeline for constructing the Op-CAD dataset. The evaluation protocol is also well specified, with separate step-level and sequence-level tasks for assessing different aspects of CAD reasoning.

2. The paper is generally well organized, and the hierarchical benchmark design makes the evaluation setting easy to understand. The dataset structure and task definitions are clearly illustrated.

3. Op-CAD provides operation-level CAD sequences with multimodal annotations, which may support research on procedural CAD generation, multimodal reasoning, and program synthesis. The benchmark also highlights current limitations of LLMs in spatial reasoning and CAD modeling.

4. The paper introduces the Op-CAD benchmark that explicitly models CAD construction as step-by-step operation sequences. The hierarchical evaluation framework, which separates step-level and sequence-level modeling, enables experiments on geometric prediction and spatial reasoning, respectively. Together with the COOP strategy, the benchmark provides a structured way to study how LLMs reason over CAD operation sequences.

Weakness:
1. COOP is mainly evaluated on the SSM task. Since the structure of COOP closely aligns with the SSM formulation, additional experiments on other CAD-related tasks would help demonstrate its general applicability.

2. The experiments primarily compare general-purpose LLMs (e.g., GPT, Gemini). However, recent work proposes models specifically designed for CAD generation, such as CAD-MLLM (Xu, Jingwei, et al. “Cad-mllm: Unifying multimodality-conditioned cad generation with mllm.” arXiv preprint arXiv:2411.04954 (2024)). Including such baselines would better position the benchmark relative to existing CAD modeling systems.

3. Most experiments focus on prompt-based evaluation. Since Op-CAD is a large-scale dataset, it would be useful to include experiments involving fine-tuning or training models on Op-CAD, which could better demonstrate its value as a training resource.

---

> ### Author Rebuttal · Authors · 2026-03-31
>
> **We sincerely thank the reviewer for the thoughtful and encouraging feedback. We greatly appreciate your recognition of the Op-CAD dataset, the hierarchical evaluation framework, and our efforts toward operation-oriented CAD modeling. Below, we address the raised weaknesses and questions in detail.**
> ***
> ### 1. Generalizability of COOP
> *(Response to “Weakness 1” and “Question 1”)*
>
> We thank the reviewer for the valuable suggestion regarding the generalizability of COOP.
>
> COOP is naturally applicable to tasks that can be decomposed into sequential modeling processes, involving **step-wise natural language guidance, long-horizon reasoning, and explicit intermediate states**. In contrast, existing CAD-related tasks (e.g., point cloud-to-CAD, image-to-CAD) are typically formulated as **one-shot** mappings without explicit intermediate states, and are therefore not directly compatible with COOP.
> We agree this is an important direction. We will include a discussion on the applicability conditions of COOP in the revised paper and explore broader extensions to other CAD-related tasks in future work.
> ***
> ### 2. Comparison with Domain-Specific CAD Models
> *(Response to “Weakness 2” and “Question 2”)*
>
> We fully agree that including domain-specific CAD models would further strengthen the evaluation. However, there are two practical limitations:
>
> 1. **Model Availability:** Recent LLM-based Text-to-CAD models (e.g., CAD-LLaMA, CAD-GPT, and CAD-MLLM) are not publicly available, which prevents direct empirical comparison.
> 2. **Architectural Mismatch:** Existing transformer-based models (e.g., Text2CAD) are primarily designed for one-shot generation from global shape descriptions, and are not directly compatible with the step-by-step, operation-oriented setting of Op-CAD.
>
> To facilitate future research and fair comparison, we will release both the Op-CAD dataset and our trained model (**Op-LLaMA-8B**).
>
> To further strengthen the evaluation, we additionally include multimodal LLMs (e.g., **Qwen3-VL-8B/30B**) on Op-CAD:
> | Model | Setting | Pass@1 ↑ | Pass@2 ↑ | Mean CD ↓ | Median CD ↓ | Mean HD ↓ | CF-IoU ↑ |
> |-------|--------|----------|----------|-----------|-------------|-----------|----------|
> | **Qwen3-VL-30B** *(vs. Qwen3-Coder-30B)* | SGM | 0.878 (**+5.3%**) | 0.909 (**-0.4%**) | 101.926 (**-1.3%**) | 65.625 (**-2.0%**) | 413.282 (**-2.4%**) | - |
> |  | SSM | 0.735 (**-4.5%**) | 0.876 (**+3.3%**) | 24.121 (**+0.8%**) | 0.926 (**-0.5%**) | 206.522 (**-1.8%**) | 0.295 (**+8.1%**) |
> | **Qwen3-VL-8B** *(vs. Qwen3-8B)* | SGM | 0.793 (**+15.3%**) | 0.893 (**+10.5%**) | 127.860 (**+0.7%**) | 87.280 (**-0.6%**) | 479.730 (**-2.9%**) | - |
> |  | SSM | 0.742 (**+16.1%**) | 0.840 (**+11.3%**) | 19.028 (**-7.4%**) | 0.786 (**0.0%**) | 200.841 (**+1.2%**) | 0.284 (**+35.2%**) |
>
> These results indicate that models with stronger visual-spatial reasoning capabilities achieve better performance, aligning with the requirements of our task.
> ***
> ### 3. Training and Fine-tuning on Op-CAD
> *(Response to “Weakness 3” and “Question 3”)*
>
> We thank the reviewer for the suggestion and for recognizing the potential of Op-CAD as a training resource.
>
> We would like to respectfully clarify that extensive fine-tuning experiments were indeed included in our original submission to explicitly demonstrate Op-CAD's effectiveness as a training resource. Specifically:
>
> 1. **LLM Fine-tuning (Op-LLaMA-8B):** As detailed in Section 5.3 and Table 5, we performed Supervised Fine-Tuning (SFT) on Llama-3.1-8B. The resulting Op-LLaMA-8B significantly outperforms the Llama-3.1-8B baseline on both SGM and SSM tasks (Pass@1 increased by 8.0% and 25.5%, respectively, and CF-IoU improved by 172.7%). It even surpasses the performance of the 8x larger Llama-3.1-70B model.
>
> 2. **Domain-Specific Training (DeepCAD-Op):** Furthermore, as shown in Section 5.3 and Table 6, we applied the DeepCAD training pipeline to our dataset. The resulting DeepCAD-Op model achieved comprehensive improvements across both DeepCAD and Op-CAD benchmarks (e.g., Mean CD decreased by 6.48% and 86.39% respectively, and ACCparam improved by 6.41%).
>
> To further highlight the training value of Op-CAD,, we conducted an **additional fine-tuning experiment** using the Qwen3-8B model. As detailed below,  the result validating that Op-CAD provides effective supervision for improving both geometric accuracy and spatial reasoning in CAD modeling.
> | Model | Setting | Pass@1 ↑ | Pass@2 ↑ | Mean CD ↓ | Median CD ↓ | Mean HD ↓ | CF-IoU ↑ |
> |------|--------|----------|----------|-----------|-------------|-----------|----------|
> | **Qwen3-8B (FT on Op-CAD)** *(vs. Qwen3-8B)* | SGM | 0.742 (**+7.8%**) | 0.861 (**+6.6%**) | 117.552 (**-7.4%**) | 75.234 (**-14.4%**) | 441.041 (**-10.7%**) | - |
> |  | SSM | 0.742 (**+16.1%**) | 0.861 (**+14.0%**) | 24.348 (**+18.5%**) | 0.489 (**-37.8%**) | 154.521 (**-22.2%**) | 0.241 (**+14.8%**) |

---

> > ### Author Rebuttal · Reviewer_tWq9 · 2026-04-03
> >
> > Thank you for the rebuttal, which resolves some of my concerns. I still have some questions:
> >
> > 1. How is the CD metric for SGM and SSM calculated? Is it obtained by averaging the CD values of $M_{step, j}$ and $M_{seq, j}$, respectively?
> >
> > 2. How can SGM and SSM be applied to evaluate performance, if the model generates a correct CAD object but from a different design history? This is likely to happen after RL training is introduced. In this case, it seems that there is no ground truth for $M_{step, j}$ and $M_{seq, j}$?
> >
> > Overall, I find Sections 3.1 and 3.2 to be reasonably innovative, and the scale and modality coverage of the Op-CAD dataset are fairly solid. However, I have some concerns about the applicability of SGM and SSM as mentioned above.
> >
> > The paper may benefit from focusing more on the following aspects:
> >
> > 1. Collecting or synthesizing more complex CAD sequences, such as Chamfer and Fillet operations, which currently remain underrepresented in the dataset.
> >
> > 2. Investigating the effectiveness of Op-CAD as a training source, for example, for CoT+RL pipelines that aim to improve reasoning capabilities, which I think is suitable for Op-CAD.

---

> > > ### Author Response · Authors · 2026-04-03
> > >
> > > **We sincerely thank the reviewer for recognizing the novelty of our work and the value of the Op-CAD dataset. We also appreciate the thoughtful follow-up questions and provide our responses below.**
> > >
> > > ---
> > >
> > > ### Q1. CD Computation
> > >
> > > Yes, your understanding is correct. For both SGM and SSM, the Chamfer Distance (CD) is computed by comparing the predicted geometry with the corresponding ground-truth geometry—at the shape of each step $M_{step,j}$ (for SGM) and on the cumulative result $M_{seq,j}$ (for SSM). We report both the mean CD and median CD across all samples for SGM and SSM.
> > >
> > > ---
> > >
> > > ### Q2. Applicability of SGM and SSM
> > >
> > > Our work addresses the ambiguity of multiple valid construction sequences in one-shot generation by enforcing a controllable, step-by-step generation process based on sequences parsed from original CAD modeling workflows, where the ground-truth geometry remains well aligned for reliable evaluation.
> > >
> > > Since SGM and SSM are geometry-oriented metrics, they focus on geometric consistency rather than enforcing a unique construction sequence; therefore, differences in construction history do not significantly affect the evaluation results.
> > >
> > > Moreover, they provide effective process-level reward signals for early-stage RL with step-by-step (curriculum) training, helping alleviate reward sparsity compared to relying solely on final-result-based evaluation, and enabling adaptation to complex CAD tasks.
> > >
> > > ---
> > >
> > > ### Q3. Complex CAD Sequences
> > >
> > > We visualize complex modeling sequences from Op-CAD and provide a representative step-by-step construction example to illustrate the complexity of our data: [https://anonymous.4open.science/r/mode-visualization/icml2026figure.pdf.](https://anonymous.4open.science/r/mode-visualization/icml2026figure.pdf) We will also follow your suggestion to further increase the proportion of Fillet and Chamfer operations in future data collection and synthesis.
> > >
> > > ---
> > >
> > > ### Q4. Op-CAD as a Training Source
> > >
> > > In this work, the effectiveness of Op-CAD as a training source is demonstrated through SFT and DeepCAD-style training, where the use of Op-CAD significantly improves performance.
> > >
> > > We will further extend this to RL and include RL-based training results in a future revision.

---

### Review · Ethics_Reviewer_zich · 2026-03-20

**Recommendation:** No remediation action needed

**Basis For Judgement:**

n/a

**Ethics Issue:**

By attempting to systematize and mirror the operations of CAD, this benchmarking exercise could cause mass layoffs of engineers who specialize in CAD.

Was the 128k operations instances in Op-CAD collected with consent of the creators?

Who "trained" Op-CAD by "decomposing [the original] CAD modeling sequences into fine-grained operations and topological relationships?"

**Remediation Action:**

Be careful of posing COOP and CAD tuning/training as naturally resulting "strategies" of your research. This seems presumptuous given the early stage of development.

---

### Decision · Program_Chairs · 2026-04-30

**Decision:**

Accept (regular)

**Comment:**

This paper presents a large-scale multi-modal dataset, OP-CAD, designed for operation-oriented CAD generation. It includes 128k operation-level instances. The paper introduces a novel evaluation metric for chamfer and fillet operations, and it presents a new prompting strategy called chain-of-operation prompting.

The paper received 4/4/3/4. The main concerns raised by Reviewer YyN2 are about the theoretical contribution rather than the engineering-driven approach. The authors convinced reviewers that the high-quality dataset is the bottleneck in the field, stating that four basic operations in OP-CAD can support 70~80% of industrial workflows. Regarding reviewer cMnX's concerns regarding formal definitions, the authors clarified the terms. Regarding reviewer tWq9's comment on the experiment with domain-specific CAD LLMs, the authors provided Qwen3-VL experiments, since CAD-MLLMs are not publicly available.

Overall, AC values the attempt to introduce a new dataset (MIT-licensed, with code and evaluation benchmarks) to the field, and AC notes that the major concerns raised by reviewers are successfully addressed. Although YyN2 expressed strong concerns about the contribution, given the problems where the dataset is a major bottleneck, AC agrees with other reviewers that this could be a good direction for future research. Therefore, AC recommends the acceptance of the paper. However, AC strongly advises authors to revise the paper based on the rebuttal materials and discussions.